# The mechanism of RNA capping by SARS-CoV-2

Gina J. Park[1,10], Adam Osinski[1,10], Genaro Hernandez[1], Jennifer L. Eitson[2], Abir Majumdar[1], Marco Tonelli[3], Katie Henzler-Wildman[3], Krzysztof Pawłowski[1,4,5], Zhe Chen[6], Yang Li[6], John W. Schoggins[2] & Vincent S. Tagliabracci[1,7,8,9 ✉]

The RNA genome of SARS-CoV-2 contains a 5′ cap that facilitates the translation of viral proteins, protection from exonucleases and evasion of the host immune response[1–4]. How this cap is made in SARS-CoV-2 is not completely understood. Here we reconstitute the N7- and 2′-O-methylated SARS-CoV-2 RNA cap ($^{7Me}GpppA_{2′-O-Me}$) using virally encoded non-structural proteins (nsps). We show that the kinase-like nidovirus RdRp-associated nucleotidyltransferase (NiRAN) domain[5] of nsp12 transfers the RNA to the amino terminus of nsp9, forming a covalent RNA–protein intermediate (a process termed RNAylation). Subsequently, the NiRAN domain transfers the RNA to GDP, forming the core cap structure GpppA-RNA. The nsp14[6] and nsp16[7] methyltransferases then add methyl groups to form functional cap structures. Structural analyses of the replication–transcription complex bound to nsp9 identified key interactions that mediate the capping reaction. Furthermore, we demonstrate in a reverse genetics system[8] that the N terminus of nsp9 and the kinase-like active-site residues in the NiRAN domain are required for successful SARS-CoV-2 replication. Collectively, our results reveal an unconventional mechanism by which SARS-CoV-2 caps its RNA genome, thus exposing a new target in the development of antivirals to treat COVID-19.

Coronaviruses (CoVs) are a family of positive-sense, single-stranded RNA viruses that cause disease in humans[9], including SARS-CoV-2, the aetiological agent of the ongoing COVID-19 pandemic[10,11]. The 5′ proximal two-thirds of the SARS-CoV-2 RNA genome contains two open reading frames that are translated by host ribosomes to form two large polyproteins[2]. These polyproteins are cleaved by viral proteases to form 16 non-structural proteins (nsp1–16), some of which make up the replication–transcription complex (RTC)[2]. At the core of the RTC is the nsp12 RNA-dependent RNA polymerase (RdRp), which is the target of antiviral agents used to treat COVID-19, including remdesivir[12] and molnupiravir[13]. nsp12 also contains an N-terminal NiRAN domain[5]. The NiRAN domain shares sequence and structural similarity with the pseudokinase selenoprotein O (SelO), which transfers AMP from ATP to protein substrates (a process termed AMPylation)[14–16]. Notably, the active-site kinase-like residues of the NiRAN domain are highly conserved in *Nidovirales* (Extended Data Fig. 1) and are required for the replication of equine arteritis virus (EAV) and SARS-CoV-1 in cell culture[5]. Several hypotheses for the function of the NiRAN domain have been proposed, including roles in protein-primed RNA synthesis, RNA ligation and mRNA capping[5,17].

The CoV RNA genome, like eukaryotic mRNAs, contains a methylated guanosine linked to the first nucleotide of the RNA through a reverse 5′ to 5′ triphosphate linkage[1,4]. Methylation of the ribose 2′-OH position of the first nucleotide completes the cap and protects the RNA from the host immune system[18,19]. This 5′ cap is important for RNA stability, initiation of mRNA translation and protection from exonucleases[20]. Thus, formation of the RNA cap is crucial for successful replication and transcription of the viral genome.

The 5′ cap structure was originally identified on RNAs of reovirus[21] and vaccinia virus[22], and the molecular mechanisms underlying the formation of the 5′ cap were subsequently determined in these systems[23,24]. This canonical capping pathway, which is conserved among most viruses as well as all eukaryotes, entails (1) an RNA triphosphatase (RTPase), which removes the γ-phosphate from the nascent 5′-triphosphorylated RNA (5′-pppRNA) to yield a 5′-diphosphorylated RNA (5′-ppRNA); (2) a guanylyltransferase (GTase), which transfers GMP from GTP to 5′-ppRNA to form the core cap structure GpppN-RNA; (3) a (guanine-N 7)-methyltransferase (N7-MTase), which methylates the cap guanine at the N7 position; and (4) a (nucleoside-2′-O)-methyltransferase (2′-O-MTase), which methylates the ribose-2′-OH position on the first nucleotide of the RNA (Extended Data Fig. 2). In CoVs, the nsp13, nsp14 and nsp16 proteins have RTPase[25], N7-MTase[6] and 2′-O-MTase[7] activities, respectively. Thus, it was presupposed that the CoV capping mechanism occurs in a similar manner to the eukaryotic capping pathway, with

[1]Department of Molecular Biology, University of Texas Southwestern Medical Center, Dallas, TX, USA. [2]Department of Microbiology, University of Texas Southwestern Medical Center, Dallas, TX, USA. [3]Department of Biochemistry, University of Wisconsin–Madison, Madison, WI, USA. [4]Department of Biochemistry and Microbiology, Institute of Biology, Warsaw University of Life Sciences, Warsaw, Poland. [5]Department of Translational Medicine, Lund University, Lund, Sweden. [6]Department of Biophysics, University of Texas Southwestern Medical Center, Dallas, TX, USA. [7]Harold C. Simmons Comprehensive Cancer Center, University of Texas Southwestern Medical Center, Dallas, TX, USA. [8]Hamon Center for Regenerative Science and Medicine, University of Texas Southwestern Medical Center, Dallas, TX, USA. [9]Howard Hughes Medical Institute, University of Texas Southwestern Medical Center, Dallas, TX, USA. [10]These authors contributed equally: Gina J. Park, Adam Osinski. ✉e-mail: vincent.tagliabracci@utsouthwestern.edu

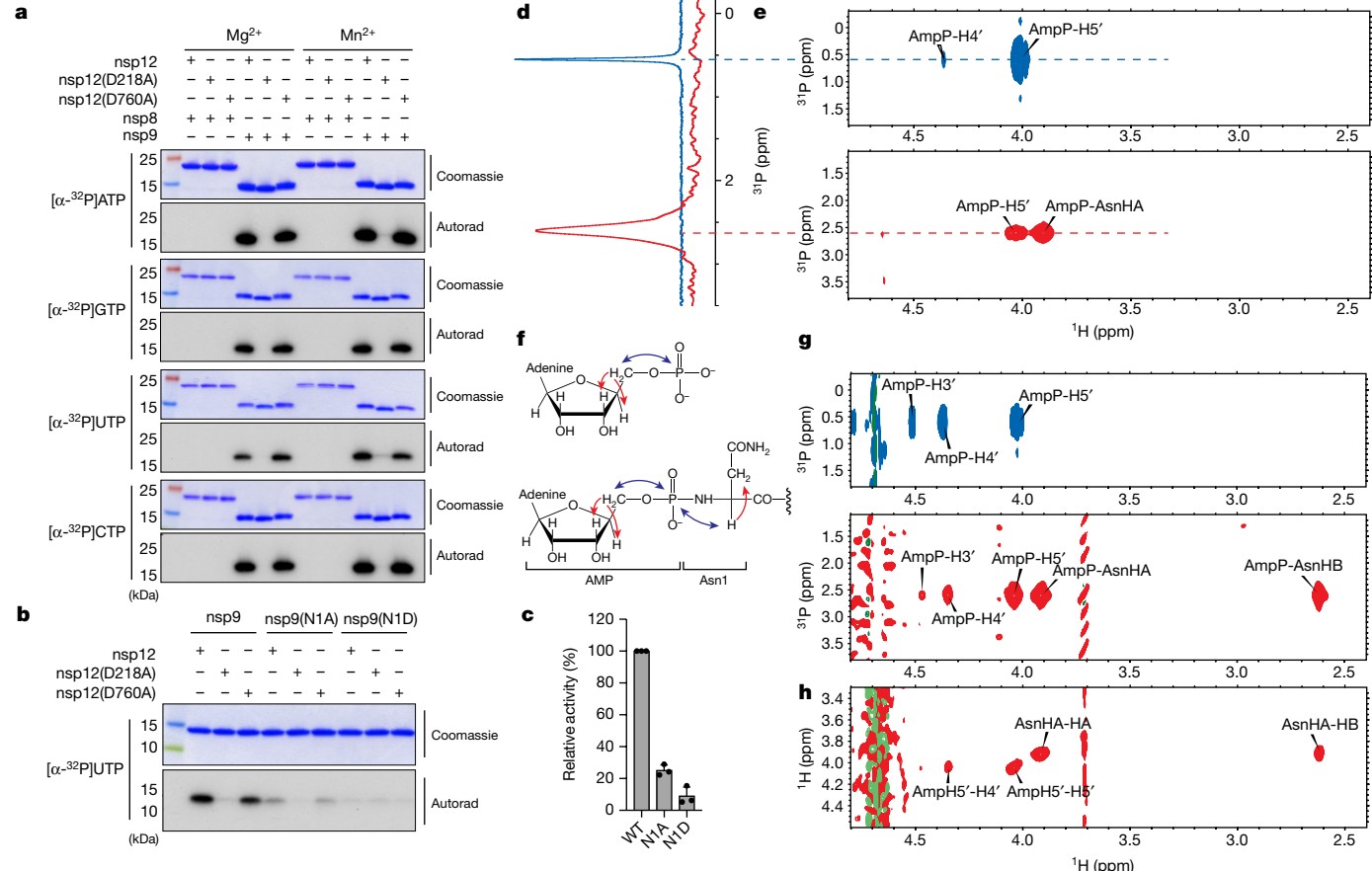

**Fig. 1 | The NiRAN domain NMPylates the N terminus of nsp9. a**, Incorporation of α-$^{32}$P from [α-$^{32}$P]ATP, GTP, UTP or CTP into nsp8 or nsp9 by WT nsp12, the NiRAN mutant (D218A) or the polymerase mutant (D760A). Reactions were performed in the presence of $Mg^{2+}$ or $Mn^{2+}$, and the products were resolved by SDS–PAGE and visualized by Coomassie staining (top) and autoradiography (autorad; bottom). **b**, Incorporation of α-$^{32}$P from [α-$^{32}$P]UTP into nsp9 or the indicated mutants by the NiRAN domain. The reaction products were analysed as described in **a**. **c**, Quantification of reaction products shown in **b** depicting the relative NiRAN-dependent UMPylation activity towards nsp9 or the indicated mutants. Radioactive gel bands were excised and quantified by scintillation counting. **d**, One-dimensional $^{31}$P

spectrum of AMP–nsp9 (red) and AMP (blue) recorded in the same buffer as the reference. **e**, 2D $^{1}$H, $^{31}$P-HSQC spectra of AMP (top, blue) and AMP–nsp9 (bottom, red). **f**, The structure of AMP (top) and AMP–nsp9 (bottom) with arrows to indicate the magnetization transfer steps that result in the peaks observed in the 2D NMR spectra. The blue arrows indicate the transfer steps that yield the peaks in the HSQC spectra, and the red arrows show the additional magnetization transfer during the TOCSY that result in the additional peaks found in the HSQC–TOCSY spectra. **g**, 2D $^{1}$H, $^{31}$P-HSQC–TOCSY spectra of AMP (top, blue) and AMP–nsp9 (bottom, red). **h**, 2D $^{1}$H, $^{1}$H-HSQC–TOCSY spectra of AMP–nsp9. For **a**–**c**, the results are representative of at least three independent experiments. For **c**, data are mean ± s.d.

the NiRAN domain functioning as the GTase[3,5,26]. However, evidence to support this claim is lacking.

Here we show that the NiRAN domain transfers monophosphorylated RNA (5′-pRNA) from 5′-pppRNA to the N terminus of nsp9 as an intermediate step in cap synthesis. The NiRAN domain then transfers 5′-pRNA from RNAylated nsp9 to GDP to form the core cap structure GpppA-RNA. We demonstrate that the N terminus of nsp9 and the kinase-like active site residues in the NiRAN domain are required for SARS-CoV-2 replication.

## The NiRAN domain NMPylates nsp9

The NiRAN domain has been shown to transfer nucleotide monophosphates (NMPs) from nucleotide triphosphates (NTPs) (referred to as NMPylation) to protein substrates, including nsp9[17] and the nsp12 co-factors, nsp7[27] and nsp8[28]. We observed NiRAN-dependent NMPylation of native nsp9, but not native nsp7 or nsp8 (Fig. 1a and Extended Data Fig. 3). Quantification of $^{32}$P incorporation and intact mass analyses suggests near stoichiometric incorporation of NMPs into nsp9 (Extended Data Fig. 3f–j). Mutation of nsp9 Asn1 to Ala or Asp reduced NMPylation of nsp9 (Fig. 1b,c), consistent with previous research suggesting that NMPylation occurs on the backbone nitrogen of nsp9

Asn1[17]. To provide direct evidence that the amino terminus of nsp9 is NMPylated by the NiRAN domain, we performed nuclear magnetic resonance (NMR) spectroscopy of AMPylated nsp9. The two-dimensional (2D) $^{1}$H, $^{31}$P heteronuclear single quantum coherence (HSQC) and 2D HSQC–total correlation spectroscopy (HSQC–TOCSY) spectra confirmed that the AMP is attached to the nitrogen backbone atom of Asn1 through a phosphoramidate linkage (Fig. 1d–h).

## The NiRAN domain RNAylates nsp9

Given the ability of the NiRAN domain to transfer NMPs to nsp9 using NTPs as substrates, we wondered whether the NiRAN domain could also use 5′-pppRNA in a similar manner (Fig. 2a). We synthesized a 5′-pppRNA 10-mer corresponding to the first 10 bases in the leader sequence (LS10) of the SARS-CoV-2 genome (hereafter 5′-pppRNA$^{LS10}$) (Supplementary Table 1). We incubated 5′-pppRNA$^{LS10}$ with nsp9 and nsp12 and analysed the reaction products using SDS–PAGE. We observed an electrophoretic mobility shift in nsp9 that was time dependent, sensitive to RNase A treatment and required an active NiRAN domain, but not an active RdRp domain (Fig. 2b). Intact-mass analyses confirmed the incorporation of monophosphorylated RNA$^{LS10}$ (5′-pRNA$^{LS10}$) into nsp9 (Extended Data

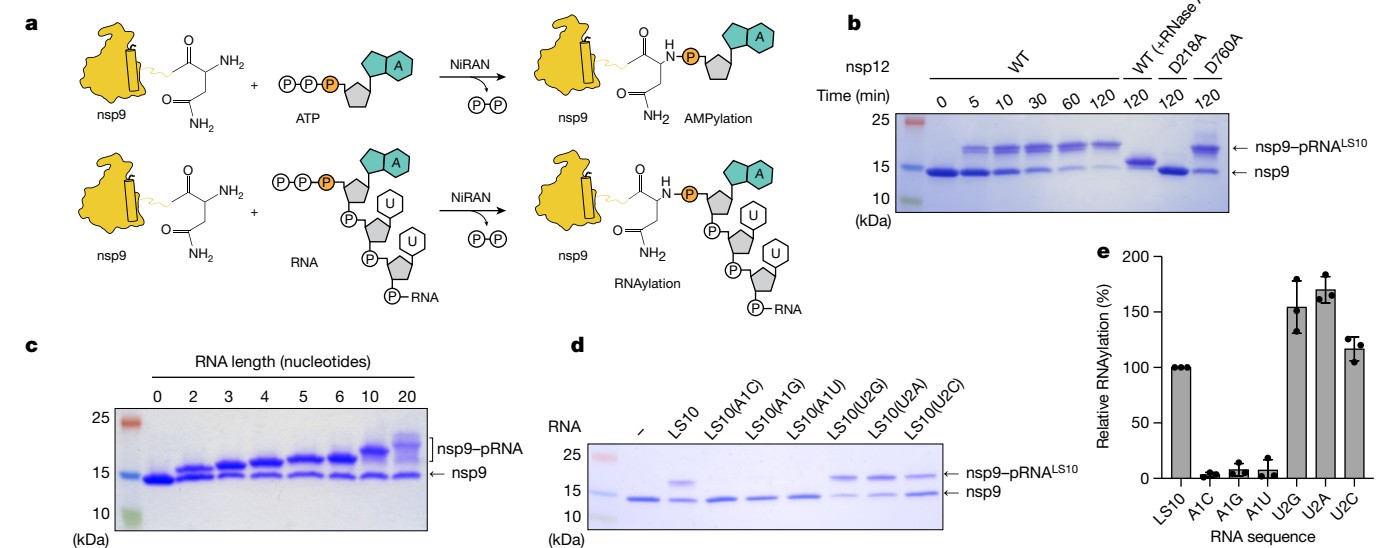

**Fig. 2 | The NiRAN domain RNAylates nsp9. a,** Schematic of the nsp9 AMPylation reaction (top) and the proposed nsp9 RNAylation reaction (bottom). **b,** Time-dependent incorporation of RNA into nsp9 by WT nsp12, the NiRAN mutant (D218A) or the polymerase mutant (D760A). The reaction products were analysed using SDS−PAGE and Coomassie staining. The samples were also treated with RNase A. **c,** Incorporation of different lengths of 5′-pppRNAs into nsp9 by the NiRAN domain. The reaction products were analysed as described in **b. d,e,** Incorporation of RNAs (with substitutions in the first and second base) into nsp9 by the NiRAN domain. Reaction products were analysed as described in **b** (**d**) and quantified by densitometry (**e**). For **e**, data are mean ± s.d. of three replicates. The results in this figure are representative of at least three independent experiments.

Fig. 4a). The reaction was dependent on $Mn^{2+}$ and required a triphosphate at the 5′-end of the RNA (Extended Data Fig. 4b,c). Substituting Ala for Asn1 reduced the incorporation of $RNA^{LS10}$ into nsp9 (Extended Data Fig. 4d). We also observed RNAylation of nsp9 using LS RNAs ranging from 2 to 20 nucleotides (Fig. 2c). Mutation of the first but not the second nucleotide markedly reduced RNAylation (Fig. 2d,e). Thus, the NiRAN domain RNAylates the N terminus of nsp9 in a substrate-selective manner.

## The NiRAN domain makes the core cap structure

Negative-sense RNA viruses of the order *Mononegavirales*, including vesicular stomatitis virus (VSV), have an unconventional capping mechanism in which a polyribonucleotidyltransferase (PRNTase) transfers 5′-pRNA from 5′-pppRNA to GDP through a covalent enzyme−RNA intermediate[29,30] (Extended Data Fig. 5a). As the NiRAN domain transfers 5′-pRNA to nsp9, we hypothesized that this protein−RNA species may be an intermediate in a similar reaction mechanism to that of the VSV system. To test this hypothesis, we purified nsp9−$pRNA^{LS10}$ and incubated it with GDP in the presence of nsp12. Treatment with GDP deRNAylated nsp9 in a NiRAN-dependent manner, as judged by the nsp9 electrophoretic mobility on SDS−PAGE (Fig. 3a) and its molecular mass based on intact-mass analysis (Extended Data Fig. 5b). The reaction was time dependent, (Fig. 3b), preferred $Mg^{2+}$ over $Mn^{2+}$ (Extended Data Fig. 5c) and was specific for GDP and−to some extent−GTP, but not the other nucleotides tested (Fig. 3c). Interestingly, although inorganic pyrophosphate ($PP_i$) was able to deAMPylate nsp9−AMP, it was unable to deRNAylate nsp9−$pRNA^{LS10}$ (see Discussion and Extended Data Fig. 5d).

We used urea−PAGE to analyse the fate of the $RNA^{LS10}$ during the deRNAylation reaction. Treatment of nsp9−$pRNA^{LS10}$ with nsp12 and [α-$^{32}$P]GDP that was generated by treatment of [α-$^{32}$P]GTP with nsp13 (Extended Data Fig. 5e) resulted in a [$^{32}$P]-labelled RNA species that migrated similarly to GpppA-$RNA^{LS10}$ produced by the vaccinia capping enzyme (VCE; Fig. 3d). The reaction was dependent on a functional NiRAN domain but not an active RdRp domain. To confirm the presence of a GpppA-RNA cap, we digested the RNA produced from the nsp12

reaction with P1 nuclease and detected GpppA using high-performance liquid chromatography−mass spectrometry (HPLC−MS) analysis (Extended Data Fig. 5f). Thus, the NiRAN domain is a GDP-PRNTase that mediates the transfer of 5′-pRNA from nsp9 to GDP.

In our attempts to generate GpppA-$RNA^{LS10}$ in a 'one pot' reaction, we found that GDP inhibited the RNAylation reaction (Extended Data Fig. 5g). However, the formation of GpppA-$RNA^{LS10}$ could be achieved in one pot provided that the RNAylation occurs before the addition of GDP (Extended Data Fig. 5g,h).

## nsp14 and nsp16 form cap-0 and cap-1 structures

The SARS-CoV-2 genome encodes an *N7*-MTase domain within nsp14[6] and a 2′-*O*-MTase in nsp16, the latter of which requires nsp10 for activity[7]. nsp14 and the nsp10−nsp16 complex use *S*-adenosyl methionine (SAM) as the methyl donor. To test whether NiRAN-synthesized GpppA-$RNA^{LS10}$ can be methylated, we incubated $^{32}$P-labelled GpppA-$RNA^{LS10}$ with nsp14 and/or the nsp10−nsp16 complex in the presence of SAM and separated the reaction products using urea−PAGE (Extended Data Fig. 6a). We extracted RNA from the reaction, treated it with P1 nuclease and CIP, and then analysed the products using thin-layer chromatography (TLC) (Fig. 4a). As expected, the NiRAN-synthesized cap migrated similarly to the GpppA standard and the products from the VCE reaction (compare lanes 1 and 4). Similarly, reactions that included SAM and nsp14 migrated similarly to the $^{7Me}$GpppA standard and to the products from the VCE reaction after the addition of SAM (compare lanes 2 and 6). Furthermore, treatment of $^{7Me}$GpppA-$RNA^{LS10}$, but not unmethylated GpppA-$RNA^{LS10}$, with the nsp10−nsp16 complex produced the $^{7Me}$GpppA-$_{2'-O-Me}$-RNA cap-1 structure (compare lanes 3, 8 and 9). In parallel experiments, we incubated NiRAN-synthesized GpppA-$RNA^{LS10}$ with nsp14 and/or the nsp10−nsp16 complex in the presence of [$^{14}$C]-labelled SAM ($^{14}$C on the donor methyl group) and separated the reaction products using urea−PAGE. As expected, nsp14 and the nsp10−nsp16 complex incorporated $^{14}$C into GpppA-$RNA^{LS10}$ to form the cap-0 and cap-1 structures, respectively (Extended Data Fig. 6b). Thus, the SARS-CoV-2 $^{7Me}$GpppA-$_{2'-O-Me}$-RNA capping mechanism can be reconstituted in vitro using virally encoded proteins.

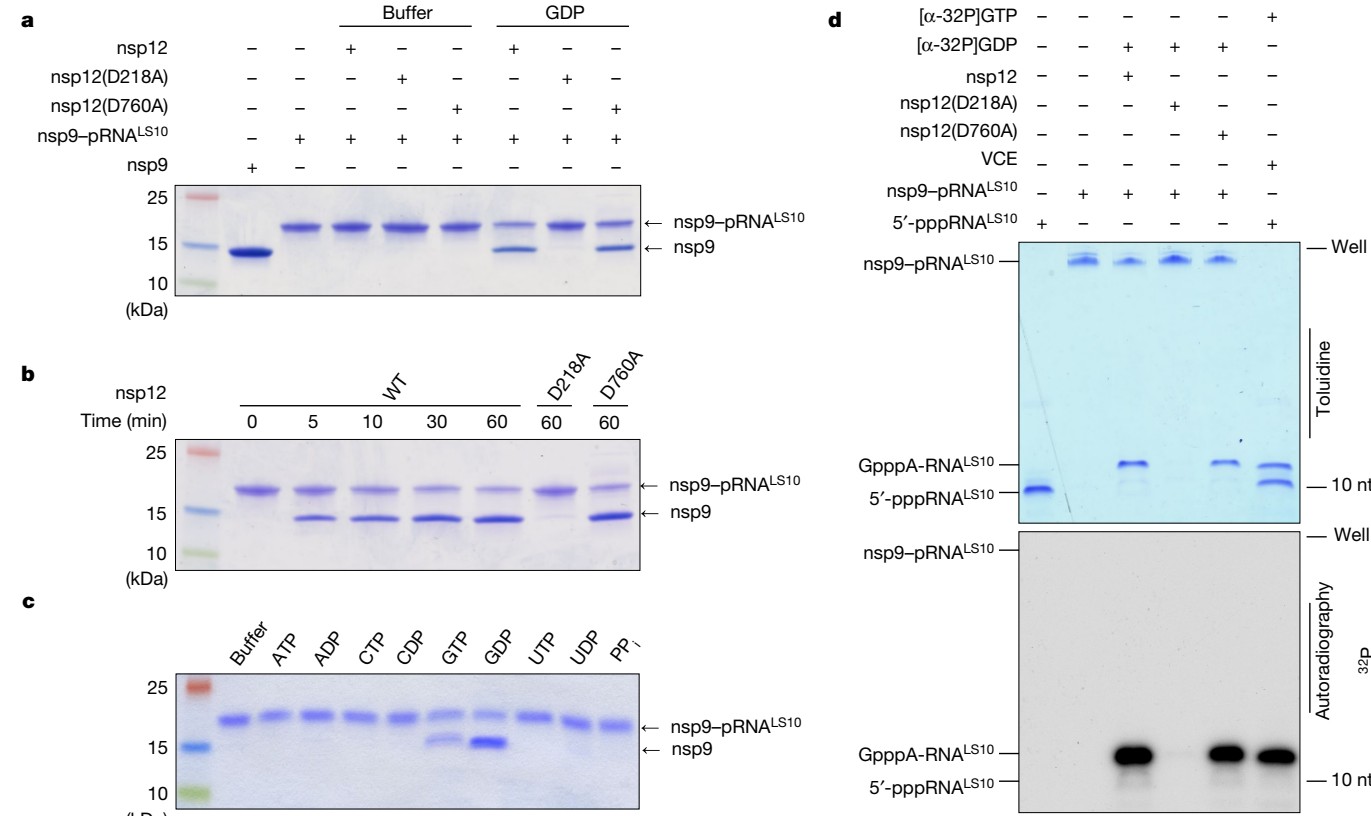

**Fig. 3 | The NiRAN domain catalyses the transfer of 5′-pRNA from nsp9 to GDP to form the core cap structure GpppA-RNA. a**, DeRNAylation of the covalent nsp9–RNA[LS10] species by WT nsp12, the NiRAN mutant (D218A) or the polymerase mutant (D760A) when incubated with buffer or GDP. The reaction products were analysed as described in Fig. 2b. **b**, Time-dependent deRNAylation of nsp9–pRNA[LS10] by WT nsp12, the NiRAN mutant (D218A) or the polymerase mutant (D760A). The reaction products were analysed as described in Fig. 2b. **c**, DeRNAylation of nsp9–pRNA[LS10] by nsp12 in the presence of different NTPs, NDPs or PP$_i$. The reaction products were analysed as described in Fig. 2b. **d**, Incorporation of α-32P from [α-32P]GDP into nsp9–pRNA[LS10] by WT nsp12, the NiRAN mutant (D218A) or the polymerase mutant (D760A). VCE was used as a control but was incubated with [α-32P]GTP. The reaction products were resolved using urea–PAGE and visualized by toluidine blue O staining (top) and autoradiography (bottom). The results in this figure are representative of at least two independent experiments. nt, nucleotides.

Efficient translation of mRNAs is dependent on eIF4E binding to the [7Me]GpppA-RNA cap[31]. To test whether the SARS-CoV-2 RNA cap is functional, we incubated [32P]-labelled [7Me]GpppA-RNA[LS10] with GST-tagged eIF4E. We observed [32P]-labelled RNA in GST pull-downs of [[32P]][7Me]GpppA-RNA but not the unmethylated derivative (Fig. 4b). Thus, the [7Me]GpppA-RNA cap generated by SARS-CoV-2-encoded proteins is a substrate for eIF4E in vitro, suggesting that the cap is functional.

## Structural insights into the capping reaction

We determined a cryo-electron microscopy (cryo-EM) structure of the nsp7–nsp8–nsp9–nsp12 complex and observed a nsp9 monomer bound in the NiRAN active site (Fig. 5a, Extended Data Fig. 7 and Supplementary Table 2). The native N terminus of nsp9 occupies a similar position to previously reported structures using a non-native N terminus of nsp9[26] (Fig. 5b,c). Our cryo-EM analysis was hindered by the preferred orientation of the complex and sample heterogeneity, yielding final maps with high levels of anisotropy, with distal portions of nsp9 missing and weak density for the N-lobe of the NiRAN domain (Extended Data Fig. 7). Thus, we used the complex structure from ref. [26] (Protein Data Bank (PDB): 7CYQ) to study the structural basis of NiRAN-mediated RNA capping.

The first four residues of nsp9 extend into the NiRAN active site, forming electrostatic and hydrophobic contacts in and around a groove near the kinase-like active site (Fig. 5d). Asn1 of nsp9 is positioned inside the active site, primed for transfer of 5′-pppRNA onto its N terminus.

Although the terminal NH$_2$ group of nsp9 is the substrate for RNAylation, the local quality of the structures is not high enough to distinguish its exact position. We modelled the nsp9 acceptor NH$_2$ pointing towards what appears to be the phosphates of the nucleotide analogue UMP-NPP in the active site (Fig. 5b). In the structure in ref. [26], Asn1 was assigned an opposite conformation and there are unmodelled residues (non-native N terminus; NH$_2$-Gly-Ser-) visible in the density maps, distorting local structural features[32] (Fig. 5c (arrow)). Nevertheless, the presence of these additional residues does not affect the overall binding mode of nsp9 to the NiRAN domain.

Asn2 of nsp9 is in a negatively charged cleft around the NiRAN active site, and contacts Arg733, which extends from the polymerase domain and positions the N terminus of nsp9 (Fig. 5e). Alanine substitution of Arg733 and an nsp12 mutant lacking the RdRp domain (ΔRdRp; 1–326) markedly reduced both RNAylation and GDP-PRNTase activities (Fig. 5f,g and Extended Data Fig. 8a,b). Leu4 and the C-terminal helix of nsp9 form hydrophobic interactions with a β-sheet (β9–β8–β10) in the N-lobe of the NiRAN domain (Fig. 5e,h). The N-terminal cap of the nsp9 C-terminal helix also forms electrostatic interactions with a negatively charged pocket on the surface of the NiRAN domain (Fig. 5h). Removal of the C-terminal helix from nsp9 (ΔC; 1–92) and Ala substitutions of Asn1 and Asn2 abolished RNAylation by nsp12 (Fig. 5i).

The NiRAN domain resembles SelO, with a root-mean-squared deviation of 5.7 Å over 224 Cα atoms (PDB: 6EAC)[14] (Extended Data Fig. 8c). Lys73 (PKA nomenclature, Lys72) forms a salt bridge with Glu83 (PKA, Glu91) from the αC (α2) helix and contacts the phosphates of the

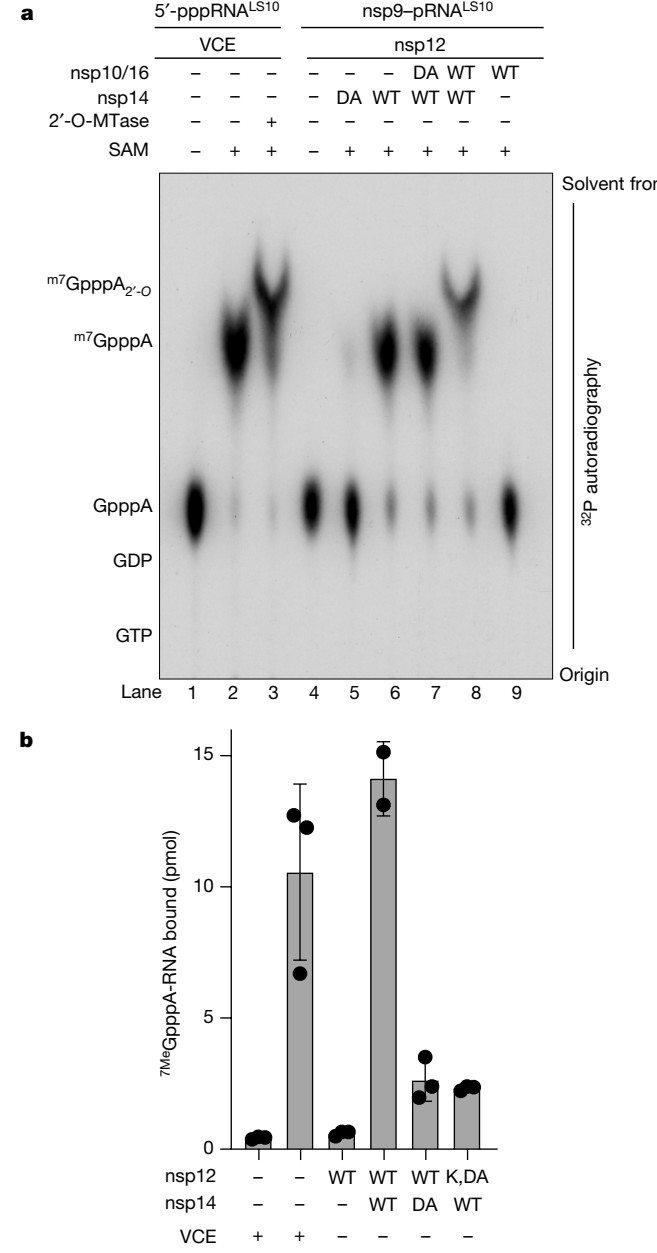

**a**

| | 5'-pppRNA^LS10 | | | nsp9–pRNA^LS10 | | | | | |
|---|---|---|---|---|---|---|---|---|---|
| | VCE | | | nsp12 | | | | | |
| nsp10/16 | − | − | − | − | − | − | DA | WT | WT |
| nsp14 | − | − | − | − | DA | WT | WT | WT | − |
| 2'-O-MTase | − | − | + | − | − | − | − | − | − |
| SAM | − | + | + | − | + | + | + | + | + |

Solvent front

$^{m7}$GpppA$_{2'-O}$

$^{m7}$GpppA

$^{32}$P autoradiography

GpppA

GDP

GTP

Origin

Lane 1 2 3 4 5 6 7 8 9

**b**

$^{7Me}$GpppA-RNA bound (pmol)

| nsp12 | − | − | WT | WT | WT | K,DA |
| nsp14 | − | − | − | WT | DA | WT |
| VCE | + | + | − | − | − | − |
| SAM | − | + | − | + | + | + |

**Fig. 4 | nsp14 and nsp16 catalyse the formation of the cap-0 and cap-1 structures. a**, TLC analysis of the reaction products from Extended Data Fig. 6a, after extraction from the urea–PAGE gel and treatment with P1 nuclease and CIP. The location of the cold standards (left) was visualized by ultraviolet fluorescence and the reaction products were visualized by $^{32}$P autoradiography. **b**, Pull-down assays depicting the binding of [$^{32}$P]$^{7Me}$GpppA-RNA^LS10 to GST–eIF4E. [$^{32}$P]$^{7Me}$GpppA-RNA^LS10 was produced using SARS-CoV-2 virally encoded proteins or the VCE. Radioactivity in GST pull-down assays was quantified by scintillation counting. Data are mean ± s.d. The results shown in the figure are representative of three independent experiments. nsp14 DA, nsp14(D331A); nsp16 DA, nsp16(D130A).

nucleotide (Fig. 5j). As expected, the 'DFG' motif Asp218 (PKA, Asp184) binds to a divalent cation. Interestingly, the NiRAN domain lacks the catalytic Asp (Extended Data Fig. 1), (PKA, Asp166); however, like in SelO, Asp208 is next to the metal-binding Asn209 (PKA, Asn171) and may act as a catalytic base to activate the NH$_2$ group on the N terminus of nsp9 (Fig. 5j). Alanine substitutions in these residues largely reduced both RNAylation (Fig. 5k) and GDP-PRNTase (Fig. 5l) activities.

In canonical kinases, the β1–β2 G-loop stabilizes the phosphates of ATP[33]. The NiRAN domain contains a β-hairpin insert (β2–β3) where the β1–β2 G-loop should be (Extended Data Fig. 8d). This insertion not only makes contacts with the N terminus of nsp9, but also contains a conserved lysine residue (Lys50) that stabilizes the phosphates of the bound nucleotide. Similarly, Arg116 also contacts the phosphates of the nucleotide. SelO contains a similar set of basic residues that accommodate the flipped orientation of the nucleotide to facilitate AMPylation (Extended Data Fig. 8d). Both Lys50 and Arg116 are required for RNAylation (Fig. 5k) and GDP-PRNTase (Fig. 5l) activities. Notably, Lys73, Arg116 and Asp218 in SARS-CoV-1 nsp12 are required for viral replication[5].

## NiRAN capping is required for viral replication

To determine the importance of the NiRAN domain and the N terminus of nsp9 in viral replication, we used a DNA-based reverse genetics system that can rescue infectious SARS-CoV-2 (Wuhan-Hu-1/2019 isolate) expressing a fluorescent reporter[8] (Extended Data Fig. 9a). We introduced single point mutations in nsp9 (N1A, N1D and N2A) and nsp12 (K73A, D218A and D760A) and quantified viral titres by plaque assay. Notably, we could not detect plaque formation in cells incubated with undiluted supernatants for any of the mutants (Fig. 5m). We also quantified the virus in supernatants of producer cells using quantitative PCR with reverse transcription (RT–qPCR) to detect the viral *N* gene and observed more than a 99% reduction in viral load for all of the mutants compared with the wild type (WT) (Extended Data Fig. 9b). To account for the possibility of a proteolytic defect in the mutant viral polyprotein, we tested whether the main viral protease nsp5 (M$^{Pro}$) can cleave a nsp8–nsp9 fusion protein containing the Asn1/Asn2 mutations in nsp9. The N1D mutant was not cleaved by nsp5, suggesting that the replication defect observed for this mutant is a result of inefficient processing of the viral polyprotein. However, the N1A and N2A mutants were efficiently cleaved by nsp5 (Extended Data Fig. 9c,d). Collectively, these data provide genetic evidence that the residues involved in capping the SARS-CoV-2 genome are essential for viral replication.

## Discussion

We propose the following mechanism of RNA capping by CoV: during transcription, the nascent 5'-pppRNA binds to the NiRAN active site, in either a *cis* (Fig. 6a) or *trans* (Fig. 6b) manner, and 5'-pRNA is subsequently transferred to the N terminus of nsp9, forming a phosphoramidate bond (Fig. 6c (i and ii)). The nsp13 protein produces GDP from GTP, which binds to the NiRAN active site and attacks RNAylated nsp9, releasing capped RNA and regenerating unmodified nsp9 (Fig. 6c (iii and iv)). Subsequently, nsp14 and nsp16 perform sequential *N*7 and 2'-*O* methylations, forming a fully functional $^{7Me}$GpppA$_{2'-O-Me}$-RNA cap.

SARS-CoV-2 nsp12 is thought to initiate transcription/replication starting with an NTP, or a short 5'-pppRNA primer[34]. Cryo-EM structures of the RTC suggest that the double-stranded RNA product makes its way out of the RdRp active site in a straight line, supported by the nsp8 helical stalks[26,35,36]. In a *cis* capping model, the helical duplex with nascent 5'-pppRNA would then need to unwind, flex 90° and extend into the NiRAN active site around 70 Å away (Fig. 6a). More likely, a separate RTC complex could perform capping in *trans* (Fig. 6b). Notably, ref. [37] proposed that the nascent RNA strand is separated from the template after passing through the proof-reading ExoN domain of nsp14 on a neighbouring RTC and threaded towards the NiRAN domain.

SARS-CoV-1 nsp13 has RNA helicase, nucleotide triphosphatase (NTPase) and RNA 5'-triphosphatase (RTPase) activities[25]. The RTPase activity implicated nsp13 in the first step of the capping mechanism; however, although nsp13 can act on 5'-pppRNA, this reaction is inhibited in the presence of cellular concentrations of ATP[25]. Thus, we favour the idea that the physiological functions for nsp13 are: (1) to use the energy from ATP hydrolysis to unwind double-stranded RNA (helicase), and (2)

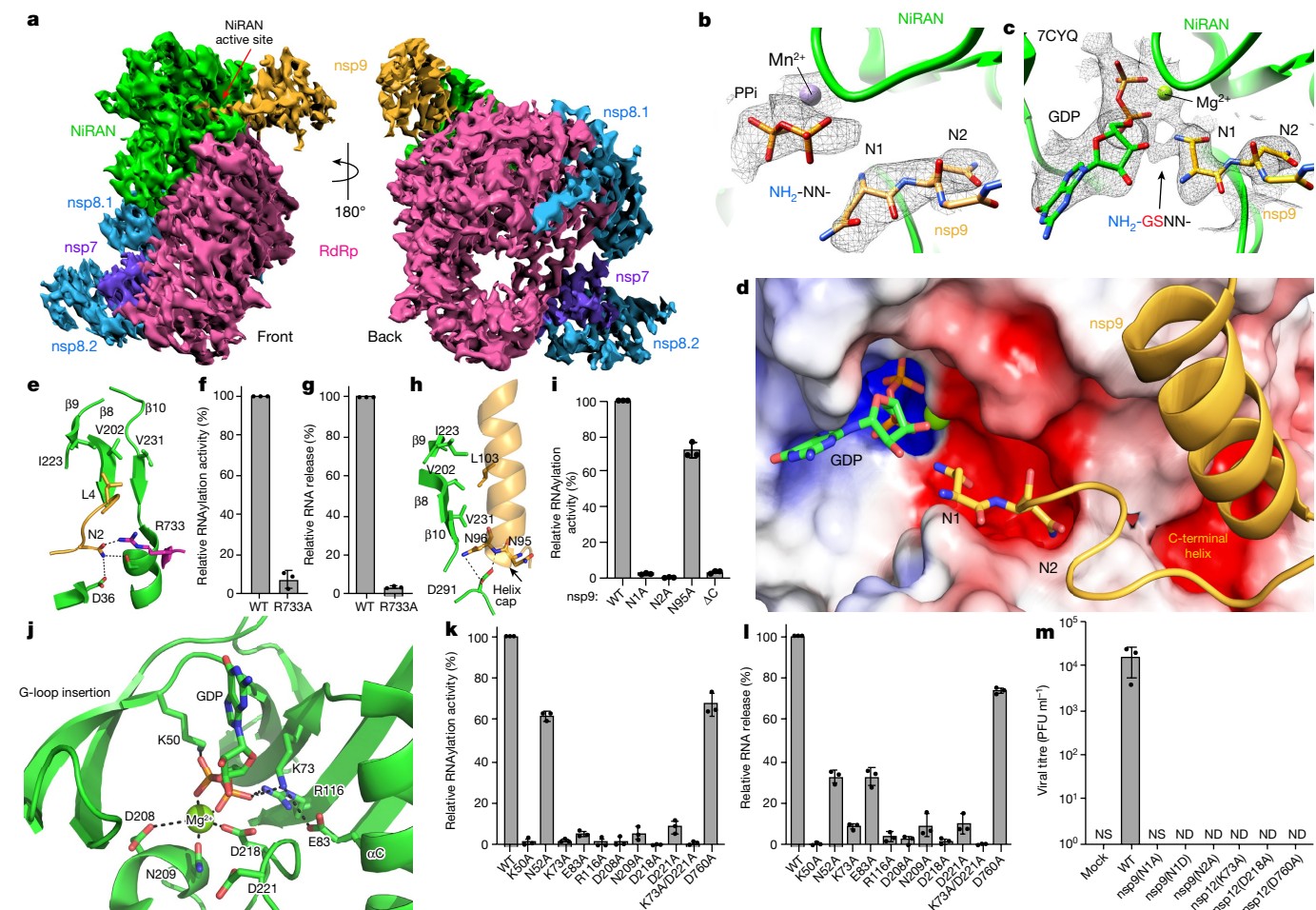

**Fig. 5 | Structural and genetic insights into RNA capping by the kinase domain. a**, Front and back views of nsp12, nsp7, nsp8 and nsp9 cryo-EM maps. Green, NiRAN; magenta, RdRp; violet, nsp7; light blue, nsp8; gold, nsp9. **b,c**, Density maps of the N terminus of nsp9 from this study (**b**) and from ref.[26] (PDB: 7CYQ) (**c**). The arrow in **c** indicates probable density of unmodelled Gly–Ser residues on non-native nsp9. **d**, Electrostatic surface of the NiRAN active site from PDB 7CYQ bound to nsp9, contoured at ±5 kT e⁻¹. **e**, Cartoon representation of the nsp9 N terminus interacting with the β9–β8–β10 sheet in NiRAN, and Arg733 in RdRp (PDB: 7CYQ). **f**, Relative RNAylation activity of the nsp12(R733A) mutant. The reaction products were analysed as described in Fig. 2e. **g**, Relative GDP-PRNTase activity of the nsp12(R733A) mutant (measured as RNA release from nsp9–pRNA^LS10). The reaction products were separated by urea–PAGE, stained with SYBR Gold and quantified by densitometry. **h**, Cartoon representation of the nsp9 C-terminal helix interacting with the NiRAN domain (PDB: 7CYQ). **i**, Relative RNAylation activity of nsp12 towards indicated nsp9 mutants. ΔC, nsp9^1–92. The reaction products were analysed as described in Fig. 2e. **j**, Cartoon representation of the NiRAN active site (PDB: 7CYQ). Catalytic residues and GDP are shown as sticks, Mg²⁺ is shown as a green sphere and interactions are denoted by dashed lines. **k**, The relative RNAylation activity of nsp12 and the indicated mutants. The reaction products were analysed as described in Fig. 2e. **l**, The relative GDP-PRNTase activity of nsp12 and the indicated mutants. The reaction products were analysed as described in **g**. **m**, SARS-CoV-2 (and the indicated mutants) viral titres quantified as plaque-forming units in the culture medium (PFU per ml). Data are mean ± s.d. of *n* = 3 biological replicates. ND, not detected. For **f**, **g**, **k** and **l**, data are mean ± s.d. of one biological replicate.

to hydrolyse GTP to GDP, which can then act as an acceptor for 5′-pRNA in the capping reaction.

The SARS-CoV-2 capping mechanism is reminiscent of the capping mechanism used by VSV, although there are some differences. The VSV large (L) protein is a multifunctional enzyme that carries out RdRp, PRNTase and methyltransferase activities to form the cap[29,38,39]. During the reaction, 5′-pRNA is transferred to a conserved His residue within the PRNTase domain, which is distinct from the protein kinases[30]. The presence of two different enzymatic mechanisms of capping, proceeding through covalent protein–RNA intermediates, in *Mononegavirales* and in *Nidovirales* is an example of convergent evolution.

Consistent with other reports[17,32], we observed NiRAN-catalysed NMPylation of nsp9 (Fig. 1a and Extended Data Fig. 3) and did not detect NMPylation of nsp7 and nsp8. Yan et al.[26] most likely did not observe NMPylation of nsp9 due to the presence of additional non-native residues at the N terminus of nsp9, which masked the modification site.

Although our results do not necessarily preclude a biologically relevant function for nsp9 NMPylation, it is worth noting that this modification is reversible in the presence of PP_i[32] (Extended Data Fig. 5d). PP_i is produced during the RdRp reaction, making the stability of NMPylated nsp9 difficult to envision in vivo. RNAylated nsp9 was not reversible in the presence of PP_i. Thus, RNAylation is probably the physiologically relevant modification of nsp9 during viral RNA capping.

Both GDP and GTP were able to deRNAylate nsp9–pRNA^LS10 (Fig. 3c) and both were able to form a GpppA cap structure in the presence of nsp9–pRNA^LS10 and nsp12 (Extended Data Fig. 10a). However, the GTP used in these assays contained GDP, probably due to GTP hydrolysis (Extended Data Fig. 10b). Thus, the NiRAN-mediated capping reaction is specific for GDP.

Recent research suggested that the NiRAN domain is a GTase that transfers GMP from GTP to 5′-ppRNA, forming a GpppA-RNA cap intermediate[3,26]. In our efforts to reproduce these results, we did not

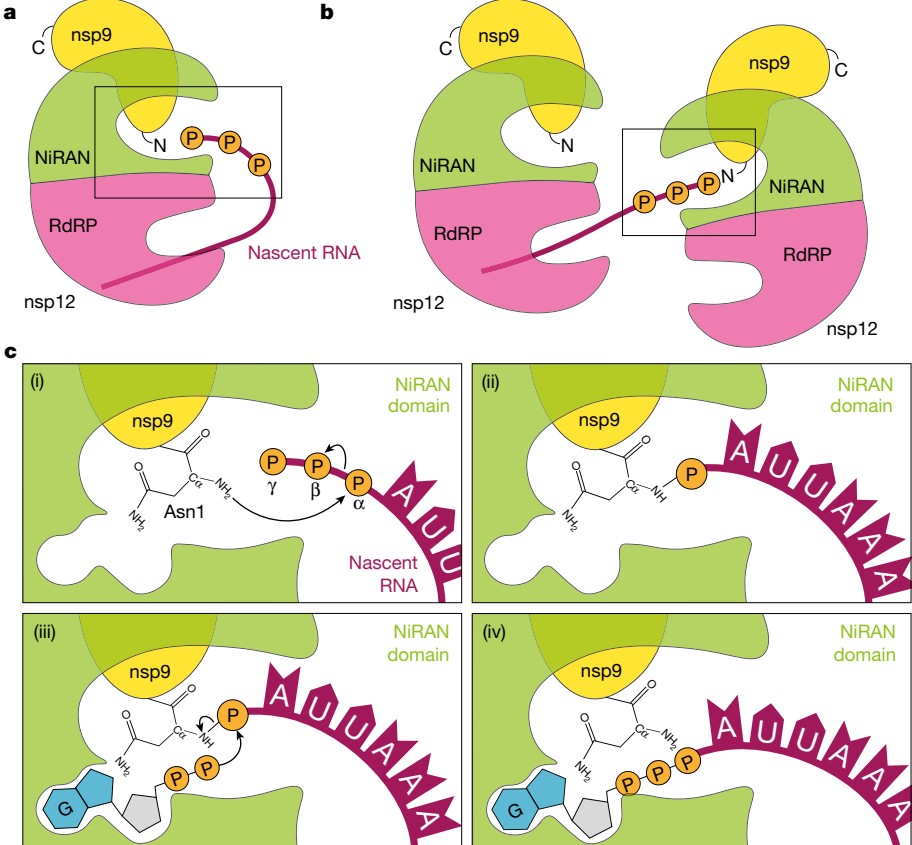

**Fig. 6 | Proposed model of the SARS-CoV-2 RNA-capping mechanism. a,b**, During transcription, the nascent 5'-pppRNA binds to the NiRAN active site in either a *cis* (**a**) or *trans* (**b**) manner. **c**, After binding, the N terminus of nsp9 attacks the α-phosphate of the nascent 5'-pppRNA (i), forming the covalent nsp9–pRNA species and releasing PP_i (ii). After GDP binding to the NiRAN active site, the β-phosphate of the nsp13-generated GDP attacks the 5'-phosphate on the nsp9–pRNA (iii), releasing capped RNA and regenerating unmodified nsp9 (iv). Subsequent methylation events are carried out by nsp14 and nsp16 to generate the $^{7Me}GpppA_{2'-O-Me}$-RNA cap.

detect nsp12-dependent GpppA cap formation by TLC or by urea–PAGE analysis of the RNA (Extended Data Fig. 10c,d), in contrast to our control, in which the VCE efficiently generated GpppA-RNA. As nsp13 and the VCE can hydrolyse GTP to GDP[25], the cap reported previously[26] appears to be GDP formed from nsp13- and VCE-dependent hydrolysis of GTP.

In summary, we have defined the mechanism by which SARS-CoV-2 caps its genome and have reconstituted this reaction in vitro using non-structural proteins encoded by SARS-CoV-2. Our results uncover new targets for the development of antiviral agents to treat COVID-19 and highlight the catalytic adaptability of the kinase domain.

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

# Methods

## Chemicals and reagents

Ampicillin sodium (A9518), ATP (A2383), ADP (A2754), chloramphenicol (C0378), CTP (C1506), CDP (C9755), dithiothreitol (DTT; D0632), EDTA (E5134), GTP (G8877), GDP (G7127), imidazole (I2399), IPTG (I5502), kanamycin sulfate (K1377), 2-mercaptoethanol (BME, M3148), Brilliant Blue R (B0149), magnesium chloride (MgCl$_2$; M2670), manganese (II) chloride tetrahydrate (MnCl$_2$; M3634), PEI-cellulose TLC plates (Z122882), potassium chloride (KCl; P9541), pyrophosphate (221368), urea (U6504), UTP (U6625) and UDP (94330) were obtained from Millipore-Sigma. Q5 DNA polymerase (M0492L), all restriction enzymes used for cloning, proteinase K (P8107S), yeast inorganic pyrophosphatase (M2403), Quick CIP (M0525S), nuclease P1 (M0660S), Vaccinia Capping System (M2080S), mRNA cap 2′-O-methyltransferase (M0366S), G(5′)ppp(5′)A RNA cap structure analogue (GpppA; S1406L) and m7G(5′)ppp(5′)A RNA cap structure analogue (m7GpppA; S1405S) were all obtained from New England Biolabs. Acetic acid (A38-212), RNase inhibitor (N8080119), 2× TBE-urea Sample Buffer (LC6876) and isopropanol (42383) were all obtained from Thermo Fisher Scientific. [α-$^{32}$P]ATP (BLU003H250UC), [α-$^{32}$P]CTP (BLU008H250UC), [α-$^{32}$P]GTP (BLU006H250UC), [α-$^{32}$P]UTP (BLU007H250UC) and S-[methyl-$^{14}$C]-adenosyl-L-methionine (NEC363010UC) were all obtained from PerkinElmer. All 5′-triphosphorylated RNAs were custom synthesized by ChemGenes. Phenylmethylsulfonyl fluoride (PMSF; 97064-898) was obtained from VWR. 4–20% Mini-PROTEAN TGX Stain-Free Protein Gels (4568096) were obtained from Bio-Rad. Uridine-5′-[(α,β)-imido] triphosphate (UMP-NPP; NU-930L) was obtained from Sapphire North America.

## Plasmids

SARS-CoV-2 *nsp7*, *nsp8*, *nsp12*, *nsp13*, *nsp14* and *nsp16* coding sequences (CDSs) were codon-optimized for bacterial expression and synthesized as gBlocks (Integrative DNA Technologies). The CDSs for *nsp9* and *nsp10* were amplified from mammalian expression vectors (a gift from N. Krogan)[40]. The CDSs were cloned into modified pET28a bacterial expression vectors containing N-terminal 6/8/10×His tags followed by the yeast Sumo (smt3) CDS amino acid mutations were introduced using QuikChange site-directed mutagenesis. In brief, primers were designed using the Agilent QuikChange primer design program to generate the desired mutation and were used in PCR reactions with PfuTurbo DNA polymerase. The reaction products were digested with Dpn1, transformed into DH5α cells and mutations were confirmed by Sanger sequencing.

For protein expression in *Escherichia coli*, ppSumo-SARS-CoV-2 nsps and mutants were cloned into a BamH1 site at the 5′ end, which introduced a Ser residue following the diGly motif in smt3. To make native N termini, the codon encoding the Ser was deleted using QuickChange mutagenesis. Thus, after cleavage with the ULP protease (after the diGly motif), the proteins contained native N termini. pGEX-2T-GST-eIF4E K119A[41] was obtained from Addgene (112818).

## Protein purification

**nsp5, nsp7, nsp8, nsp10, nsp14 and nsp16.** 6×His-Sumo-nsp5/7/8/10/14/16 and the corresponding mutant plasmids (with native N termini following the diGly motif in Sumo) were transformed into Rosetta (DE3) *E. coli* or LOBSTR-BL21(DE3)-RIL cells under 50 µg ml$^{-1}$ kanamycin exposure. LB Miller growth medium starter cultures (5 ml) containing 50 µg ml$^{-1}$ kanamycin and 34 µg ml$^{-1}$ chloramphenicol were grown for 2–4 h at 37 °C and then transferred to growth medium containing the same antibiotics. Typically, 2 l were grown for each protein. Protein expression was induced at an optical density (OD) of 0.7–1.0 by adding 0.4 mM IPTG and overnight incubation (16 h) at 18 °C. Cultures were centrifuged at 3,000g for 10 min and the bacterial pellet was resuspended in lysis buffer (50 mM Tris pH 8.0, 300 mM NaCl, 17.4 µg ml$^{-1}$ PMSF, 15 mM imidazole pH 8.0 and 5 mM β-ME) and lysed by sonication. Lysates were centrifuged for 30 min at 30,000–35,000g and the supernatants incubated with Ni-NTA resin for 1–2 h at 4 °C. The Ni-NTA resin was washed with 50 mM Tris pH 8.0, 300 mM NaCl, 30 mM imidazole pH 8.0, 1 mM DTT and the protein was eluted in 50 mM Tris pH 8.0, 300 mM NaCl, 300 mM imidazole pH 8.0 and 1 mM DTT. The eluted proteins were incubated with 5 µg ml$^{-1}$ Ulp1 protease overnight at 4 °C. nsp7 was separated from 6×His−Sumo by anion exchange (Capto HiRes Q 5/50 column (Cytiva) equilibrated in 50 mM Tris pH 8.0, 50 mM NaCl, 1 mM DTT, eluted with 0–50% gradient of buffer containing 1 M NaCl). 6×His−SUMO−nsp8 was treated with Ulp1 on the Ni-NTA resin, to separate nsp8 and 6×His−SUMO in buffer with no imidazole. Proteins were further purified by size-exclusion chromatography using the Superdex 200 10/300 increase, Superdex 200 16/600, Superdex 75 10/300 increase or Superdex 75 16/600 column in 50 mM Tris pH 7.5–8.0, 150–300 mM NaCl and 1 mM DTT, depending on yield and size. The fractions containing proteins of interest were pooled, concentrated in an Amicon Utra-15 centrifugal filters (pore size, 3–50 kDa). The nsp16 protein was incubated with Ni-NTA resin post SEC to remove 6×His−SUMO. Purified proteins were aliquoted and stored at −80 °C.

**nsp9.** 6×His-Sumo-nsp9 and respective mutant plasmids (with native N termini following the diGly motif in Sumo) were transformed into Rosetta (DE3) *E. coli* cells under 50 µg ml$^{-1}$ kanamycin exposure. LB Miller growth medium starter cultures (5 ml) containing 50 µg ml$^{-1}$ kanamycin and 34 µg ml$^{-1}$ chloramphenicol were grown for 2–4 h at 37 °C and then transferred to Terrific Broth (TB) growth medium containing the same antibiotics and several drops of Antifoam B emulsion (Sigma-Aldrich, A5757). Protein expression was induced at an OD of 1.2 by adding 0.4 mM IPTG and incubating overnight (for 16 h) at 18 °C. The cultures were centrifuged at 3,000g for 10 min and the bacterial pellet was resuspended in lysis buffer (50 mM Tris pH 8.0, 300 mM NaCl, 10% glycerol 17.4 µg ml$^{-1}$ PMSF, 15 mM imidazole pH 8.0 and 5 mM β-ME) and lysed by sonication. Lysates were centrifuged for 30 min at 30,000g and the supernatants incubated in Ni-NTA resin for 1–2 h at 4 °C. The Ni-NTA resin was washed with 50 mM Tris pH 8.0, 300 mM NaCl, 10% glycerol, 30 mM imidazole pH 8.0 and 1 mM DTT, and each protein was eluted in 50 mM Tris pH 8.0, 300 mM NaCl, 10% glycerol, 300 mM imidazole pH 8.0 and 1 mM DTT. The eluted protein was incubated with 5 µg ml$^{-1}$ Ulp1 overnight at 4 °C. Proteins were diluted with 50 mM Tris pH 8.0, 10% glycerol, 1 mM DTT buffer to lower the NaCl concentration to 30 mM and subsequently ran through a Hi-Trap CaptoQ column where the flowthrough contained purified nsp9. NaCl was added to each protein to a final concentration of 150 mM, concentrated in an Amicon Ultra-15 with a 10 kDa molecular weight cut-off (MWCO), aliquoted and stored at −80 °C.

**nsp12.** 8×His or 10×His-Sumo−nsp12 and respective mutant plasmids (with native N termini following the diGly motif in Sumo) were transformed into LOBSTR-BL21(DE3)-RIL *E. coli* cells under 50 µg ml$^{-1}$ kanamycin exposure. LB Miller growth medium starter cultures (5 ml) containing 50 µg ml$^{-1}$ kanamycin and 34 µg ml$^{-1}$ chloramphenicol were grown for 2–4 h at 37 °C and then transferred to 1 l growth medium containing the same antibiotics. Protein expression was induced at an OD of 0.8–1.2 by adding 0.4 mM IPTG and incubating overnight (for 16 h) at 18 °C. The cultures were centrifuged at 3,000–3,500g for 10 min and the bacterial pellet was resuspended in lysis buffer (50 mM Tris pH 8.0, 300 mM NaCl, 10% glycerol, 17.4 µg ml$^{-1}$ PMSF, 15 mM imidazole pH 8.0 and 5 mM β-ME) and lysed by sonication. The lysates were centrifuged for 30 min at 30,000–35,000g and the supernatants were incubated in Ni-NTA resin for 1–2 h at 4 °C. The Ni-NTA resin was washed with high-salt buffer (50 mM Tris pH 8.0, 1 M NaCl, 10% glycerol, 30 mM imidazole and 5 mM β-ME), followed by a high-imidazole wash (50 mM Tris pH 8.0, 300 mM NaCl, 10% glycerol, 75 mM imidazole and 5 mM β-ME), and the protein was eluted in 50 mM Tris pH 8.0, 300 mM

NaCl, 10% glycerol, 300 mM imidazole pH 8.0 and 1 mM DTT. The eluted proteins were incubated with 5 µg ml$^{-1}$ Ulp1 overnight at 4 °C. Proteins were further purified by size-exclusion chromatography using the Superdex 200 10/300 increase column, or Superdex 200 16/600 in 50 mM Tris pH 8.0, 150–300 mM NaCl and 1 mM DTT. The fractions containing nsp12 were pooled, concentrated in an Amicon Ultra-15 with a 30–50 kDa MWCO centrifugal filter, aliquoted and stored at −80 °C.

**nsp13.** 6×His-Sumo-nsp13 (used in Extended Data Fig. 10c) or 10×His-Sumo-nsp13 (used for [α$^{32}$P]GTP conversion into GDP) and respective mutant plasmids (with native N termini following the diGly motif in Sumo) were transformed into Rosetta (DE3) *E. coli* cells under 50 µg ml$^{-1}$ kanamycin exposure. LB Miller growth medium starter cultures (5 ml) containing 50 µg ml$^{-1}$ kanamycin and 34 µg ml$^{-1}$ chloramphenicol were grown for 2–4 h at 37 °C and then transferred to 1 l of growth medium containing the same antibiotics. Protein expression was induced at an OD of 1.0 by adding 0.4 mM IPTG and overnight incubation (16 h) at 18 °C. The cultures were centrifuged at 3,000–3,500*g* for 10 min and the bacterial pellet was resuspended in lysis buffer (50 mM Tris pH 8.0, 300 mM NaCl, 10% glycerol, 17.4 µg ml$^{-1}$ PMSF, 15 mM imidazole pH 8.0 and 5 mM β-ME) and lysed by sonication. The lysates were centrifuged for 30 min at 30,000–35,000*g* and the supernatants were incubated in Ni-NTA resin for 1–2 h at 4 °C. Ni-NTA resin for 6×His–Sumo–nsp13 was washed with 50 mM Tris pH 8.0, 300 mM NaCl, 10% glycerol, 30 mM imidazole pH 8.0 and 1 mM DTT. 10×His–Sumo–nsp13 was washed with high-salt buffer (50 mM Tris pH 8.0, 1 M NaCl, 10% glycerol, 30 mM imidazole and 5 mM β-ME), followed by a high-imidazole wash (50 mM Tris pH 8.0, 300 mM NaCl, 10% glycerol, 75 mM imidazole and 5 mM β-ME), and the protein was eluted in 50 mM Tris pH 8.0, 300 mM NaCl, 10% glycerol, 300 mM imidazole pH 8.0 and 1 mM DTT. The eluted protein was incubated with 5 µg ml$^{-1}$ Ulp1 overnight at 4 °C. Proteins were buffer-exchanged or dialysed into a buffer containing 50 mM Bis-Tris pH 6.0, 30 mM NaCl, 10% glycerol and 1 mM DTT, and ion-exchange chromatography was then performed in a 5/50 MonoS column. The fractions containing nsp13 were pooled and further purified by size-exclusion chromatography using the Superdex 200 10/300 increase or Superdex 200 16/600 column in 50 mM Tris pH 8.0, 150 mM NaCl, 10% glycerol (only in 6×His), 1 mM DTT. The fractions containing nsp13 were pooled, concentrated in an Amicon Ultra-15 with a 30–50 kDa MWCO, aliquoted and stored at −80 °C.

**eIF4E.** For the production of GST–eIF4E(K119A), LOBSTR-BL21(DE3)-RIL cells were transformed with pGEX-2T-GST-eIF4E(K119A)[41] and were grown in LB supplemented with 100 µg l$^{-1}$ ampicillin. Protein expression was induced at an OD of 0.6 by adding 0.4 mM IPTG and incubating overnight (for 16 h) at 25 °C. The cultures were centrifuged at 3,000–3,500*g* for 10 min and the bacterial pellet was resuspended in lysis buffer (50 mM Tris pH 8.0, 300 mM NaCl, 17.4 µg ml$^{-1}$ PMSF and 5 mM β-ME) and lysed by sonication. Lysates were centrifuged for 30 min at 35,000*g* and the supernatants incubated with Pierce Glutathione resin for 1–2 h at 4 °C. The resin was washed with lysis buffer, and the GST–eIF4E(K119A) eluted with 50 mM Tris-HCl pH 8.0, 300 mM NaCl, 10 mM glutathione, 1 mM DTT. The protein was purified using size-exclusion chromatography on the Superdex 200 16/600 column in 50 mM Tris-HCl pH 8.0, 300 mM NaCl and 1 mM DTT, concentrated and stored as described above.

**Ipp1.** For the production of yeast inorganic pyrophosphatase (ipp1), The *S. cerevisiae ipp1* CDS was cloned into pProEx2 containing an N-terminal 6×His-TEV linker and was transformed into Rosetta *E. coli* cells under 100 µg ml$^{-1}$ ampicillin exposure. LB Miller growth medium starter cultures (5 ml) containing 100 µg ml$^{-1}$ ampicillin were grown for 2–4 h at 37 °C and then transferred to 1 l growth medium containing the same antibiotics. Protein expression was induced at an OD of 0.8–1.2 by adding 0.4 mM IPTG and incubating (for 16 h) at 18 °C. The cultures

were centrifuged at 3,000–3,500*g* for 10 min and the bacterial pellet was resuspended in lysis buffer (50 mM Tris pH 8.0, 300 mM NaCl, 17.4 µg ml$^{-1}$ PMSF, 15 mM imidazole pH 8.0 and 5 mM β-ME) and lysed by sonication. The lysates were centrifuged for 30 min at 35,000*g* and the supernatants incubated in Ni-NTA resin for 1 h at 4 °C. Ni-NTA resin was washed with high-salt buffer (50 mM Tris pH 8.0, 1 M NaCl, 30 mM imidazole and 5 mM β-ME), and was eluted in 50 mM Tris pH 8.0, 50 mM NaCl, 300 mM imidazole pH 8.0 and 1 mM DTT. The eluted protein was loaded onto the Capto HiRes Q 5/50 column (Cytiva) equilibrated in 50 mM Tris pH 8.0, 50 mM NaCl and 1 mM DTT, eluted with a 0–50% gradient of buffer containing 1 M NaCl. Protein was further purified by size-exclusion chromatography using the Superdex 200 16/600 column in 25 mM Tris pH 7.5, 50 mM NaCl and 2 mM DTT. The fractions containing YIPP were pooled, concentrated and stored as described above.

**nsp8–nsp9 fusion.** The 6×His-Sumo-nsp8-nsp9 plasmid and mutants (N1A, N1D and N2A) were transformed into Rosetta (DE3) *E. coli*. Cells were grown in Terrific Broth medium in the presence of 50 µg ml$^{-1}$ kanamycin and 25 µg ml$^{-1}$ chloramphenicol to an OD of 1.0 and induced with 0.4 mM IPTG for 16 h at 18 °C. The cultures were centrifuged at 3,500*g* for 15 min, and the pellets were resuspended in lysis buffer (50 mM Tris, pH 8.0, 500 mM NaCl, 25 mM imidazole and 10% glycerol) in the presence of 1 mM PMSF. Cells were lysed by sonication and the lysates were cleared by centrifugation at 25,000*g* for 1 h. The lysate was passed over Ni-NTA beads, which were washed with lysis buffer. The protein samples were eluted with elution buffer (50 mM Tris, pH 8.0, 300 mM NaCl, 300 mM imidazole, 5% glycerol) and cleaved overnight at 4 °C with Ulp Sumo protease. The protein samples were further purified into cleavage assay buffer (50 mM Tris, pH 7.4, 150 mM NaCl, 5% glycerol) by size-exclusion chromatography using the Superdex 75 Increase 10/300 GL column.

### GTase-activity assays

GTase-activity assays were performed as described previously[26]. Reactions were assembled in 20 µl containing 50 mM Tris pH 8.0, 5 mM KCl, 1 mM DTT, 0.005 U ml$^{-1}$ inorganic pyrophosphatase, 10 µM 5′-pppACCCCCCCCCCCCCCCCCCCCC-3′ (5′-pppRNA$^{A19C}$), 1.25 mM RNase inhibitor and, where indicated, 0.5 µM nsp12, nsp12(D218A), nsp13, nsp13(K288A) or 1 U ml$^{-1}$ of VCE. The reactions were started with 1 mM MgCl$_2$, 100 µM [α-$^{32}$P]GTP (specific radioactivity = 1,000 cpm pmol$^{-1}$) and incubated for 1 h at 37 °C. Half of the reaction was stopped by adding 0.8 U ml$^{-1}$ proteinase K and was incubated for 30 min at 37 °C before the addition of 2× RNA loading dye (Novex) and incubated for 3 min at 95 °C. The reaction products were resolved in a 15% TBE-urea–PAGE gel. The gel was then stained with toluidine blue O and the $^{32}$P signal was detected by autoradiography.

The other half of the GTase reactions were treated with 10 U ml$^{-1}$ P1 nuclease for 1 h at 37 °C. The reactions were then split in half again with one half treated with 1 U ml$^{-1}$ Quick CIP for 30 min at 37 °C. The reactions were spotted on a PEI cellulose TLC plate and developed in a 0.4 M ammonium sulfate (NH$_4$)$_2$SO$_4$ solvent system. The plate was dried and the $^{32}$P signal was detected by autoradiography.

### NMPylation assays

NMPylation reactions were carried out in 20 µl containing 50 mM Tris (pH 7.5), 5 mM KCl, 1 mM DTT, 16 µM nsp7, nsp8 or nsp9 (and mutants), and 4.8 nM nsp12 (and mutants). The reactions were started with 1 mM MgCl$_2$ or MnCl$_2$, 200 µM [α-$^{32}$P]ATP, [α-$^{32}$P]UTP, [α-$^{32}$P]GTP or [α-$^{32}$P]CTP (specific radioactivity = 1,000 cpm pmol$^{-1}$). The reactions were incubated at 37 °C for 5 min and stopped by adding 2 µl of 500 mM EDTA, followed by addition of 5× SDS–PAGE sample buffer with 10% β-ME and incubated for 3 min at 95 °C. The reaction products were resolved by SDS–PAGE on a 4–20% gradient gel and visualized by staining with Coomassie Brilliant Blue. The $^{32}$P signal was detected by autoradiography and scintillation counting.

For quantification of stoichiometry for nsp7, nsp8 and nsp9, [α-$^{32}$P] UTP was used and the reactions were allowed to proceed for 16 h at 37 °C and processed as above.

## nsp9 NMPylation kinetics

NMPylation reactions were carried out in a 20 µl reaction containing 50 mM Tris (pH 7.5), 5 mM KCl, 1 mM DTT, 16 µM nsp9, and 4.8 nM nsp12. The reactions were started by adding MnCl$_2$ and [α-$^{32}$P]ATP, CTP, GTP or UTP as indicated. The final concentration in the reaction was 0.5–200 µM (specific radioactivity of around 5,000 cpm pmol$^{-1}$) of the indicated nucleotide triphosphate and 1 mM MnCl$_2$. The reactions were incubated at 37 °C for 5 min and stopped by adding 2 µl of 500 mM EDTA, followed by the addition of 5× SDS–PAGE sample buffer + β-ME and boiling for 2–5 min. The reaction products were resolved by SDS–PAGE on a 4–20% gradient gel and visualized by staining with Coomassie Brilliant Blue. Incorporation of $^{32}$P was quantified by excising the nsp9 bands from the gel and scintillation counting. Background radioactivity was subtracted from each measurement. Rate measurements were fit to Michaelis–Menten kinetic models and $K_m$ and $V_{max}$ values for substrates were calculated by nonlinear regression using Prism v.9.3.0 for macOS (GraphPad Software; www.graphpad.com).

## NMR

For NMR studies, non-isotopically enriched AMPylated nsp9 was dissolved in 50 mM Tris buffer at pH 7.5, 150 mM NaCl, 1 mM DTT and 10% D$_2$O for spectrometer locking. The final protein concentration of this solution was 0.5 mM. A total volume of 500 µl was then used with a 5 mm NMR tube to record all of the spectra.

All of the NMR experiments were run on the Bruker Avance III spectrometer operating at 600 MHz (1H) and equipped with a 5 mm proton-optimized quadruple resonance cryogenic probe. The temperature of the sample was regulated at 308 K throughout data collection.

A one-dimensional (1D) $^{31}$P spectrum was recorded with 8,192 scans and a repetition delay of 1.5 s for a total collection time of 3.5 h. The $^{31}$P spectral window and offset were set to 17 ppm and 2.6 ppm, respectively. Waltz16 decoupling was used on $^1$H during $^{31}$P acquisition.

To observe contacts between $^{31}$P and the nearest $^1$H nuclei, a 2D $^1$H, $^{31}$P-HSQC spectrum was recorded with 1,024 and 22 complex points in the direct $^1$H and indirect $^{31}$P dimensions, respectively. The spectral window and offset were set to 16.7 ppm and 4.7 ppm for the $^1$H dimension and 3.4 ppm and 2.6 ppm for the $^{31}$P dimension, respectively. Each FID was accumulated with 1,536 scans with a repetition delay of 1 s for a total recording time of approximately 21 h.

A 2D $^1$H, $^{31}$P-HSQC–TOCSY spectrum was recorded using spectral window and offset parameters for the $^1$H and $^{31}$P dimensions similar to the HSQC spectrum described above. A 60 ms long $^1$H-$^1$H TOCSY pulse train using a DIPSI-2 sequence and a field strength of 10 kHz was tagged at the end of the HSQC sequence to observe signals from $^1$H nuclei that were further away from $^{31}$P. Given the lower sensitivity of this experiment, each FID was accumulated with 4,096 scans and a repetition delay of 1 s was used for a total recording time of 2 days and 14 h.

Using a similar pulse sequence, a 2D $^1$H,$^1$H-HSQC–TOCSY spectrum was also recorded by evolving the indirect $^1$H dimension instead of $^{31}$P. The spectral window for the 1H indirect dimension was set to 4.2 ppm, whereas the offset was maintained at 4.7 ppm as for the direct $^1$H dimension. Forty complex points were recorded for then indirect $^1$H dimension, using 2,048 accumulations for each FID and a repetition delay of 1 s for a total recording time of 2 days and 14 h.

All 2D spectra were processed using NMRPipe[42] and analysed using NMRFAM-SPARKY[43].

## Intact-mass analysis

Protein samples were analysed by LC–MS, using a Sciex X500B Q-TOF mass spectrometer coupled to an Agilent 1290 Infinity II HPLC. Samples were injected onto a POROS R1 reverse-phase column (2.1 mm × 30 mm, 20 µm particle size, 4,000 Å pore size) and desalted. The mobile phase flow rate was 300 µl min$^{-1}$ and the gradient was as follows: 0–3 min, 0% B; 3–4 min, 0–15% B; 4–16 min, 15–55% B; 16–16.1 min, 55–80% B; 16.1–18 min, 80% B. The column was then re-equilibrated at the initial conditions before the subsequent injection. Buffer A contained 0.1% formic acid in water and buffer B contained 0.1% formic acid in acetonitrile.

The mass spectrometer was controlled by Sciex OS v.1.6.1 using the following settings: ion source gas 1, 30 psi; ion source gas 2, 30 psi; curtain gas, 35; CAD gas, 7; temperature, 300 °C; spray voltage, 5,500 V; declustering potential, 80 V; collision energy, 10 V. Data were acquired from 400–2,000 Da with a 0.5 s accumulation time and 4 time bins summed. The acquired mass spectra for the proteins of interest were deconvoluted using BioPharmaView v.3.0.1 (Sciex) to obtain the molecular mass values. The peak threshold was set to ≥5%, reconstruction processing was set to 20 iterations with a signal-to-noise threshold of ≥ 20 and a resolution of 2,500.

## RNAylation assays

RNAylation reactions were typically carried out in a 10 µl volume containing 50 mM Tris (pH 7.5), 5 mM KCl, 1 mM DTT, 0.4 µg yeast inorganic pyrophosphatase, 20 µM nsp9 and 1 µM nsp12. The reactions were started by adding MnCl$_2$ and 5′-pppRNA$^{LS10}$ to a final concentration of 1 mM and 100 µM, respectively. The reactions were incubated at 37 °C for the indicated time points and stopped by addition of 5× SDS-PAGE sample buffer + β-ME and boiling the samples for 5 min. The reaction products were resolved by SDS–PAGE on a 4–20% gradient gel and visualized by Coomassie staining and quantified with Fiji[44] where indicated.

For the reactions testing RNA length specificity (Fig. 2c), 7.5 µl of a reaction master mix containing nsp9, nsp12, and yeast inorganic pyrophosphatase was added to 2.5 µl of start mix consisting of MnCl$_2$ and the indicated 5′-pppRNA. The final reaction conditions were as follows: 50 mM Tris pH 7.5, 5 mM KCl, 1 mM DTT, 0.4 µg yeast inorganic pyrophosphatase, 20 µM nsp9, 2 µM nsp12, 1 mM MnCl$_2$ and 100 µM of the indicated RNA. The reactions were incubated for 30 min at 37 °C, then stopped by addition of 5× SDS–PAGE sample buffer + β-ME and boiling the samples for 5 min. The reaction products were resolved by SDS–PAGE on a 4–20% gradient gel and visualized by Coomassie staining.

For RNAylation reactions comparing different RNA sequences (Fig. 2d,e) or nsp9 mutants (Fig. 5i), reactions were performed as described above, except using 1 µM nsp12. The reactions were incubated for 5 min and stopped by addition of 5× SDS–PAGE sample buffer + β-ME and boiling the samples for 5 min. The reaction products were resolved by SDS–PAGE on a 4–20% gradient gel and visualized by Coomassie staining.

For nsp12-mutant RNAylation comparisons, reactions contained 0.5 µM nsp12 WT or the indicated mutants and were conducted at room temperature for 60 min. Triplicate reactions were stopped with 5× SDS–PAGE loading dye, resolved on 15% SDS–PAGE gels, stained with Coomassie Brilliant Blue R and quantified with Fiji[44].

## Purification of nsp9–pRNA$^{LS10}$ species

Purified native nsp9 (0.8 mg ml$^{-1}$, 65 µM) was incubated at room temperature overnight with 130 µM of 5′-pppRNA$^{LS10}$ and ~0.8 µM of nsp12 in presence of 0.05 mg ml$^{-1}$ yeast inorganic pyrophosphatase and 1 mM MnCl$_2$, in the reaction buffer (50 mM Tris pH 7.5, 5 mM KCl, 1 mM DTT). The samples were clarified by centrifugation to remove any precipitate and applied directly onto a Capto HiRes Q 5/50 column (Cytiva) equilibrated in 50 mM Tris pH 8.0, 50 mM NaCl and 1 mM DTT. An elution gradient of 0–50% with 1 M NaCl was applied over 30 column volumes. Under these conditions, RNA and nsp9–pRNA$^{LS10}$ bound to the column and unmodified nsp9 did not. nsp9–pRNA$^{LS10}$ and unreacted RNA$^{LS10}$ eluted as a peak doublet around 70 mS cm$^{-1}$. The fractions were pooled, and further purified over Superdex 75 increase 10/300 GL (50 mM Tris pH 8.0, 300 mM NaCl, 1 mM DTT), separating

nsp9–pRNA$^{LS10}$ from unreacted RNA$^{LS10}$ and nsp12. nsp9–pRNA$^{LS10}$ was quantified by spectrophotometry with an estimated extinction coefficient of $\epsilon_{260} = 130{,}650$ M$^{-1}$ cm$^{-1}$. We also generated nsp9–pRNA$^{LS10}$ in the absence of inorganic pyrophosphatase. This nsp9–pRNA$^{LS10}$ was used in control reactions to test for PP$_i$ hydrolysis, necessary to ensure that the PPi-mediated deRNAylation reactions shown in Extended Data Fig. 5d were not affected by pyrophosphate hydrolysis. The results were similar to those presented in the figure, therefore confirming that there was no contaminating inorganic pyrophosphatase in the assays.

## DeRNAylation of nsp9–pRNA$^{LS10}$

DeRNAylation reactions were typically performed in a 10 μl reaction volume consisting of 50 mM Tris pH 7.5, 5 mM KCl, 1 mM DTT, 20 μM nsp9–pRNA$^{LS10}$, 500 nM nsp12, 1 mM MgCl$_2$ and 500 μM GDP. The reactions were started by adding MgCl$_2$/GDP and incubated at 37 °C for 5–60 min as indicated. The reactions were stopped by addition of 5× SDS–PAGE sample buffer + β-ME and boiling the samples for 5 min. The reaction products were resolved by SDS–PAGE on a 4–20% gradient gel and visualized with Coomassie staining.

For nsp12 mutant deRNAylation comparison, reactions contained 0.5 μM nsp12 WT or mutants and were conducted at room temperature for 45 min. Triplicate reactions were stopped with 2× formamide loading dye, resolved on 15% (19:1) TBE urea–PAGE gels, stained with SYBR Gold (Thermo Fisher Scientific) and quantified with Fiji[44].

For deRNAylation reactions comparing various NTPs and NDPs, reactions were performed in 10 μl volume consisting of 50 mM Tris pH 7.5, 5 mM KCl, 1 mM DTT, 20 μM nsp9–pRNA$^{LS10}$, 500 nM nsp12, 1 mM MgCl$_2$ and 500 μM of the indicated NTP or NDP. The reactions were incubated for 5 min at 37 °C and stopped by addition of 5× SDS–PAGE sample buffer with β-ME and boiling for 5 min. The reaction products were resolved by SDS–PAGE on a 4–20% gradient gel and visualized with Coomassie staining.

## Generation of [α-$^{32}$P]GDP using nsp13

To generate [α-$^{32}$P]GDP from [α-$^{32}$P]GTP, 0.3–1 mM of [α-$^{32}$P]GTP (specific activity ~2,000 cpm pmol$^{-1}$) was incubated with 0.5–1 mg ml$^{-1}$ nsp13 (and in some cases yeast cet1 NTPase) in 20 μl reaction buffer (depending on the amount needed) consisting of 50 mM Tris (pH 7.5), 5 mM KCl, 1 mM DTT and 2 mM MgCl$_2$. The reactions were started by addition of enzyme and allowed to proceed for 30 min at 37 °C. After incubation for 30 min, the reactions were boiled at 95 °C for 5 min to inactivate the nsp13 or cet1.

To compare [α-$^{32}$P]GDP and [α-$^{32}$P]GTP in RNA capping reactions (Extended Data Fig. 10a), [α-$^{32}$P]GTP was treated as described above, except without the addition of nsp13.

## Generation of radiolabelled GpppA-RNA$^{LS10}$ from nsp9–pRNA$^{LS10}$ and [α-$^{32}$P]GDP

Reactions were performed in a 10 μl volume containing 50 mM Tris pH 7.5, 5 mM KCl, 1 mM DTT, 15 μM nsp9–pRNA$^{LS10}$, 377 nM nsp12, 1 mM MgCl$_2$ and 500 μM [α-$^{32}$P]GDP (specific radioactivity of around 2,000 cpm pmol$^{-1}$). The reactions were started by addition of [α-$^{32}$P]GDP/MgCl$_2$ mixture (generated as described above) and incubated for 30 min at 37 °C. As a control, VCE was used but with [α-$^{32}$P]GTP. VCE assays were generally performed as described in the NEB Capping Protocol (M2080) with the following modifications: the reaction contained 20 μM of 5′-pppRNA$^{LS10}$, 500 μM [α-$^{32}$P]GTP (specific activity of around 2,000 cpm pmol$^{-1}$), and did not contain SAM. The reactions were stopped by the addition of 2× TBE-urea sample buffer, boiled for 5 min and resolved by urea–PAGE (20%).

## nsp13 ATPase and GTPase malachite green kinetics

nsp13 triphosphatase reactions were carried out in a 30 μl reaction volume containing 50 mM Tris (pH 7.5), 5 mM KCl, 1 mM DTT and 30 nM nsp13. The reactions were started by adding ATP/MgCl$_2$ or GTP/MgCl$_2$ as indicated. The final concentrations in the reactions were 8–500 μM of the indicated nucleotide triphosphate and 1 mM MgCl$_2$. The reactions were incubated at room temperature for 10 min and stopped by adding 30 μl of malachite green reagent. Malachite green absorbance was measured with a Spectramax ID3 spectrometer (Molecular Devices). Background activity was subtracted from each measurement and a phosphate standard curve was generated with KH$_2$PO$_4$ to determine the amount of inorganic phosphate generated. Rate measurements were fit to Michaelis–Menten kinetic models and $K_m$ and $V_{max}$ values for substrates were calculated by nonlinear regression using Prism v.9.3.0 for macOS (GraphPad Software).

## GDP inhibition of RNAylation (one-pot capping assays)

Reactions were performed in 50 mM Tris pH 7.5, 5 mM KCl, 1 mM DTT, 1 mM MgCl$_2$ and 1 mM MnCl$_2$, and contained 20 μM nsp9, 2 μM nsp12 and 100 μM 5′-pppRNA$^{LS10}$ in a volume of 20 μl. The [α-$^{32}$P]GDP was prepared as described above (using 400 μM GTP, [α-$^{32}$P]GTP at a specific activity of around 1,500 cpm pmol$^{-1}$) and diluted to final reaction concentrations in the range of 6.25–100 μM. The reactions were started by the addition of nsp12. GDP was added either before the addition of nsp12 ($t = 0$), or after 30 min of preincubation. After an additional 30 min, the reactions were split in half and stopped by the addition of 5× SDS-PAGE or 2× formamide loading dyes and the products were analysed by 4–20% gradient SDS–PAGE gel, or 15% 19:1 TBE urea–PAGE gel, respectively.

## LC–MS/MS analysis of GpppA

DeRNAylation reactions (in triplicate) were performed in 20 μl of buffer solution containing 50 mM Tris pH 7.5, 5 mM KCl, 20 μM nsp9–pRNA$^{LS10}$, 2 μM WT nsp12 or the D218A mutant, 1 mM MgCl$_2$ and 100 μM GDP. The reactions were started by adding MgCl$_2$ and GDP and were allowed to proceed for 1 h at 37 °C. After incubation for 1 h, the reactions were supplemented with 2 μl of 10× P1 buffer and 1 μl of nuclease P1 enzyme and allowed to proceed for an additional 30 min at 37 °C. The reactions were stopped by boiling for 5 min and submitted for LC–MS/MS analysis.

For standards, 20 μl of blank reaction buffer (50 mM Tris pH 7.5, 5 mM KCl, 1 mM DTT) plus 40 μl blank reaction buffer containing 0.33 μM (final) m7GpppA (m7G(5′)ppp(5′)A RNA Cap Structure Analogue, New England Biolabs, S1405S) as an internal standard (IS) was spiked with varying concentrations of GpppA (New England Biolabs, S1406L). The reaction samples (20 μl) were diluted with blank reaction buffer, containing 0.33 μM (final) m7GpppA IS, at 1:2 to a total volume of 60 μl. Standards and samples were mixed with 60 μl of 100% methanol, vortexed and then centrifuged for 5 min at 16,100$g$. The supernatant was removed and analysed by LC–MS/MS using a Sciex QTRAP 6500+ mass spectrometer coupled to a Shimadzu Nexera X2 LC. GpppA was detected with the mass spectrometer in positive multiple reaction monitoring mode by following the precursor to fragment ion transition 772.9 to 604.0. A Thermo Fisher Scientific BioBasic AX column (2.1 × 50 mm, 5 μm packing) was used for chromatography with the following conditions: buffer A: 8:2 dH$_2$O:acetonitrile + 10 mM ammonium acetate, pH 6; buffer B: 7:3 dH$_2$O:acetonitrile + 1 mM ammonium acetate, pH 10.5, 0.5 ml min$^{-1}$ flow rate; 0–1 min, 0% B; 1–2.5 min, gradient to 35% B; 2.5–5 min, 35% B; 5–7 min, gradient to 65% B; 7–10 min, 65% B; 10–10.5 min, gradient to 100% B; 10.5–15 min, 100% B; 15–15.5 min, gradient to 0% B; 15.5–20.5 min, 0% B. m7GpppA (transition 787.1 to 508.0) was used as an internal standard. Peak areas were determined, and data were further analysed using the Sciex Analyst v.1.7.2 software package. Back-calculations of standard curve samples were accurate to within 15% for 100% of these samples at concentrations ranging from 0.001 μM to 10 μM. A limit of detection was defined as a level three times that observed in blank reaction buffer and the limit of quantification as the lowest point on the standard curve that gave an analyte signal above the limit of detection and within 20% of nominal after back-calculation. The limit of quantification for GpppA was 0.005 μM.

## Methyltransferase assays

In 10 µl reactions, 40 µM nsp9–pRNA$^{LS10}$ was incubated with 2 µM nsp12 in the presence of 1 mM MgCl$_2$ and 100 µM [$^{32}$P]GDP (generated as described above) for 60 min at 37 °C. The reactions were filled to 15 µl with nsp14 and SAM (final 0.05 mg ml$^{-1}$ and 100 µM, respectively), and incubated for another 30 min. Treatment with nsp10 and nsp16 can be done concurrently with nsp14; however, the nsp10–nsp14 complex partially processes RNA$^{LS10}$, resulting in a mobility shift[45]. Thus, before addition of nsp10 and nsp16, nsp14 exonuclease activity was removed by heat inactivation (5 min at 95 °C). For 2′-O methylation, the reactions were supplemented with nsp10 and nsp16 and fresh SAM to final concentrations of 0.05 mg ml$^{-1}$ nsp10, 0.05 mg ml$^{-1}$ nsp16, 100 µM SAM in a final volume of 20 µl. Vaccinia reactions were conducted according to the manufacturer's instructions. Reactions were stopped by adding 2× formamide loading dye and were separated on 20% TBE-urea polyacrylamide gels (19:1). Radioactivity was visualized by autoradiography and RNA by toluidine staining. For TLC analysis, bands with detectible $^{32}$P signal were excised, fragmented and incubated overnight at 55 °C in elution buffer (1 M ammonium acetate, 0.2% SDS and 20 mM EDTA), rotating top over bottom. The solutions were filtered using 0.22 µm centrifugal filters, supplemented with 23 µg Glyco Blue co-precipitant (Invitrogen) and precipitated for 1 h at −20 °C by addition of isopropyl alcohol to a final concentration of 60%. The pellets were washed once with 70% ethanol and reconstituted in 10 µl of P1 buffer with P1 nuclease (NEB). After 30 min at 37 °C, the reactions were supplemented with Quick CIP (NEB) and rCut Smart buffer to a final volume of 12 µl. After further incubation for 30 min, the reactions were spotted onto PEI-Cellulose F TLC plates and resolved in 0.4 M ammonium sulfate mobile phase. Beforehand, TLC plates were prepared by development in water, removing yellow discoloration. The $^{32}$P signal was detected by autoradiography, and compared with cold standards of GTP, GDP, GpppA and $^{m7}$GpppA detected by absorption of plate fluorescence, excitable with a ultraviolet lamp $\lambda = 265$ nm. The position of $^{m7}$GpppA$_{2'-O-Me}$ was determined from the VCE and vaccinia 2′-O-methyltransferase control reaction.

Reactions with $^{14}$C-labelled SAM were conducted as described above, with two differences: cold GDP was used at 100 µM (Millipore Sigma, G7127) and 55 µM [$^{14}$C]SAM (Perkin Elmer); SAM was used at the supplied radioactivity of 52.6 mCi mmol$^{-1}$ (-117 cpm pmol$^{-1}$), with no further dilution using cold SAM.

## GST–eIF4E pulldown of $^{7Me}$GpppA-RNA

Capping reactions were set up in 20 µl and contained 2 µM nsp12, 0.04 mg ml$^{-1}$ nsp14 WT or D331A, 30 µM nsp9–pRNA$^{LS10}$, 100 µM [$\alpha^{32}$P] GDP (specific radioactivity = 1,000 cpm pmol$^{-1}$), 100 µM SAM, 2 mM MgCl$_2$. The reaction buffer was 5 mM Tris pH 8.0, 5 mM KCl, 1 mM DTT. VCE controls were performed according to manufacturer's instructions, with the same nucleotide and SAM concentrations as nsp12 reactions. After incubation for 90 min at 37 °C, 15.6 µg of GST–eIF4E(K119A) was added, along with 15 µl of Glutathione resin (Pierce, Thermo Fisher Scientific). The reactions were filled to 700 µl with 50 mM Tris pH 8.0, 150 mM NaCl, 1 mM DTT and nutated for 1 h. The resin was washed three times with 500 µl of 50 mM Tris pH 8.0, 150 mM NaCl, 1 mM DTT and radioactive signal was quantified by scintillation counting.

## Cryo-EM grid preparation

To form nsp12–nsp7–nsp8 core complex (RTC), native nsp12, nsp7 and nsp8 were incubated at a 1:2:4 molar ratio and run over the Superdex 200 increase 10/300 GL column to separate unassociated monomers. The purified complex was concentrated using spin concentrators (Amicon, 10 kDa MWCO, Sigma-Millipore), and quantified by spectrophotometry. A 3× molar excess of nsp9 over RTC was added, followed by 0.05 mM final DDM detergent immediately before freezing. The final concentration of nsp12–nsp7–nsp8 was 2 mg ml$^{-1}$. The buffer contained 50 mM Tris pH 7.5, 150 mM NaCl, 1 mM DTT, 2 mM MnCl$_2$ and 1 mM UMP-NPP. Copper Quantifoil 1.2/1.3 mesh 300 grids were used to freeze 3.5 µl of sample at 100% relative humidity using Vitrobot Mark IV (Thermo Fisher Scientific).

## Cryo-EM data collection

Before data collection, sample grids were screened on the Talos Artica microscope at the Cryo Electron Microscopy Facility (CEMF) at UT Southwestern. Cryo-EM data of nsp12–nsp7–nsp8–nsp9 complex were collected on the Titan Krios microscope at the Cryo-Electron Microscopy Facility (CEMF) at UT Southwestern Medical Center, with the post-column energy filter (Gatan) and a K3 direct-detection camera (Gatan), using SerialEM[46]. A total of 4,770 videos were acquired at a pixel size of 0.55 Å in super-resolution counting mode, with an accumulated total dose of 54 e$^-$ Å$^{-2}$ over 50 frames. The defocus range of the images was set as −1.0 to −2.5 µm.

## Image processing and 3D reconstruction

Unless described otherwise, all datasets were processed with Relion[47]. Videos were aligned and summed using MotionCor2[48], with a downsampled pixel size of 1.09 Å. The CTF parameters were calculated using Gctf[49], and images with estimated CTF max resolution better than 5 Å were selected for further processing. A total of 4,196,086 particles were picked using crYOLO[50] from 4,757 images, and extracted with a rescaled pixel size of 2.19 Å. A total of 663,999 particles were selected and re-extracted after multiple rounds of 2D and 3D classifications in Relion with the original pixel size of 1.09 Å. An additional round of 3D classification was carried out, followed by particle reduction with a custom script to remove particles from dominant orientations. The remaining 89,945 particles were processed for 3D refinement, CTF refinement and particle polishing sequentially. A final round of 3D classification with a reference mask led to 39,985 particles, which were then imported into cryoSPARC[51] for one round of non-uniform refinement. The map resolution was reported at 3.18 Å from cryoSPARC with the gold standard FSC method.

## nsp5 cleavage reactions

Concentrated protein samples were diluted in cleavage buffer (50 mM Tris, pH 7.4, 150 mM NaCl, 5% glycerol) and each reaction was performed in a total volume of 10 µl. Initial experiments measured the nsp5 concentration dependence of the nsp8–nsp9 cleavage reaction. To measure the time dependence, nsp5 (2.5 µM final concentration) was added to the nsp8–nsp9 fusion protein (12.5 µM final concentration). The reactions were incubated at 37 °C for varying amounts of time (0 to 80 min) and terminated by boiling the samples for 5 min in the presence of SDS–PAGE loading buffer. The reaction products were resolved on a 4–20% gradient tris-glycine gel and the products were visualized by Coomassie staining.

## Model building and refinement

The model was built using PDB 7CYQ as a template[26]. The model was manually rebuilt into the map using Coot[52], and refined using Phenix real space refinement[53]. Model validation was performed using MolProbity software[54].

## Bioinformatics

A representative subset of NiRAN domain sequences, provided as a multiple alignment in the original NiRAN publication[5], was supplemented by additional sequences used in this study (SARS-CoV-2, OC43, 229E strains). The human SELO sequence was added according to a FATCAT structural alignment[55] between SARS-CoV-2 nsp12 and bacterial SELO (PDB: 7CYQ and 6EAC). The alignment was visualized using the ESPript server[56].

## SARS-CoV-2 infection experiments

**Plasmid construction.** To generate recombinant SARS-CoV-2 expressing ZsGreen mutants, the infectious clone pCC1-4K-SARS-

CoV-2-Wuhan-Hu-1-ZsGreen was used as the parental backbone[8]. To generate nsp9(N1A, N1D or N2A) and nsp12(K73A) mutants, the mutations were introduced by overlap extension PCR. In brief, two fragments for each mutant were created by PCR using Herculase II Fusion DNA Polymerase (Agilent Technologies). These fragments shared homology at the 3′ end of fragment 1 and the 5′ end of fragment 2. The resulting 2 fragments were then used as a template for a third fragment to PCR a full-length amplicon containing the mutation flanked by Pac1 and MluI sites on the ends. Mutations were confirmed by DNA sequencing. To generate nsp12 mutants (D218A or D760A), a SARS-CoV-2 shuttle vector (ps1180.SARS-CoV-2-shuttle) was created using the ps1180.delXhoISacII plasmid as the backbone. Using Gibson cloning, a restriction enzyme linker that contained unique restriction sites specific to the SARS-CoV-2 genome was inserted. Smaller fragments of the SARS-CoV-2 genome were digested from the pCC1-4K-SARS-CoV-2-Wuhan-Hu-1-ZsGreen plasmid and ligated into ps1180.SARS-Cov-2-shuttle plasmid to create three new plasmids (MluI/SacI fragment for D218A region; SacI/Bsu36I fragment for D760A region). gBlocks containing mutations were synthesized by IDT and introduced into the SARS-CoV-2 shuttle vector by Gibson Assembly following standard protocols. To reassemble the full-length parental pCC1-4K-SARS-CoV-2-Wuhan-Hu-1-ZsGreen containing the new mutants, pCC1-4K-SARS-CoV-2-Wuhan-Hu-1-ZsGreen was digested with the PacI/MluI, MluI/SacI and SacI/Bsu36I restriction enzymes. An approximately 28–35 kb fragment was purified for each digest. PCR amplicons ($nsp9^{N1A}$, $nsp9^{N1D}$, $nsp9^{N2A}$, $nsp12^{K73A}$) were digested with PacI/MluI and the 5.3 kb fragment was purified. The SARS-CoV-2 shuttle plasmids were digested as follows: $nsp12^{D218A}$ (MluI/SacI releasing a 1.4 kb fragment); $nsp12^{D760A}$ (SacI/Bsu36I/PvuI releasing 3 kb fragments). All of the fragments were purified using the QIAexII Gel Purification Kit according to the standard protocol (Qiagen). The fragments were ligated together at a 3:1 ratio overnight. Ligated DNA was precipitated using 7.5 M ammonium acetate, glycogen and isopropanol, followed by a wash with ethanol. The DNA was electroporated into TransforMAX EPI300 electrocompetent *E. coli* (Lucigen). An overlapping eight-fragment PCR strategy was used to verify individual colonies by colony PCR. Confirmed colonies were grown in 10 ml tryptic soy broth (TSB; Sigma-Aldrich) containing 12.5 μg ml$^{-1}$ chloramphenicol for 6–8 h, with shaking at 37 °C. A 10 ml culture was inoculated into 100 ml TSB/chloramphenicol culture and incubated overnight, with shaking at 37 °C. The overnight culture was diluted 1:5 into fresh TSB/chloramphenicol containing 0.1% arabinose and incubated for an additional 5 h. The bacteria were pelleted and the DNA was isolated using a homemade midi prep protocol followed by the Machery-Nagel NucleoBond Xtra Midi Kit (Thermo Fisher Scientific). Full-length infectious clone plasmid was confirmed by restriction digestion and eight-fragment PCR. A list of the oligonucleotides used is provided in Supplementary Table 3.

**Virus production.** To generate virus from DNA-based infectious clones, 3 μg of plasmid was transfected into 2 individual wells of a 6-well plate of 400,000 BHK-21J cells using X-treme Gene9 Transfection Reagent (Sigma-Aldrich). Three days after transfection, the supernatant from two individual wells was combined and 4 ml was transferred to a T25 flask containing $1 \times 10^6$ VeroE6-C1008-TMPRSS2 cells and incubated for 30 min. Inoculum was removed and 5 ml MEM supplemented with 10% FBS, 1× NEAA and 1× sodium pyruvate was added. After 4 days, 250 μl supernatant was added to 750 μl TriReagent for RNA extraction and RT–qPCR. The remaining supernatant was frozen for plaque assays. T25 flasks were fixed with 4% paraformaldehyde, imaged on the Nikon Eclipse Ti system and processed with ImageJ.

**RNA extraction and iPCR.** RNA was isolated using the Direct-zol RNA mini prep kit according to the manufacturer's instructions (ZymoResearch). A 20 μl reaction contained 5 μl RNA, 5 μl TaqMan Fast Virus 1-Step Master Mix and 1.8 μl SARS-CoV-2 primer/probe set containing 6.7 μM of each primer and 1.7 μM probe (the final concentrations of

primer and probe were 600 nM primer and 150 nM probe). SARS-CoV-2 primers and probes were designed as recommended by the Center for Disease Control (https://www.cdc.gov/coronavirus/2019-ncov/lab/rt-pcr-panel-primer-probes.html). All oligonucleotides were synthesized by LGC Biosearch Technologies. RT was performed at 50 °C for 5 min, followed by inactivation at 95 °C for 2 min, and 40 cycles of PCR (95 °C for 3 s, 60 °C for 30 s) on the QuantStudio 3 (Applied Biosystems) system.

**Cells.** BHK-21J cells (a gift from C. Rice) were grown in MEM (Gibco) supplemented with 10% FBS and 1× NEAA. VeroE6-C1008 cells (ATCC) were transduced with lentiviral vector SCRBBL-TMPRSS2, selected with blasticidin (15 μg ml$^{-1}$, Gibco) and maintained in MEM supplemented with 10% FBS, 1× NEAA, 1× sodium pyruvate and 8 μg ml$^{-1}$ blasticidin.

**Plaque assay.** Viral titre was determined by plaque assay on Vero-C1008 cells (ATCC). Cells were plated at 650,000 cells per well and infected with 3 or 6 dilutions from a tenfold dilution series in 1% FBS/1× non-essential amino acids/MEM for 60 min at 37 °C, with rocking every 15 min. Overlay medium (2 ml) consisting of a 1:1 mix of 2× modified Eagle medium (Temin's Modification) supplemented with 2 penicillin–streptomycin, 2× GlutaMAX, 10% FBS:2× Avicel RC-591 (Dupont) was added to cells. Cells were incubated for 3 days at 37 °C before the overlay medium was removed and cells were fixed with 4% paraformaldehyde (Sigma-Aldrich) for 30 min at room temperature. The fixative was removed, and cells were stained for at least 5 min with crystal violet (0.2% crystal violet (Sigma-Aldrich), 20% ethanol), before removal and plaque enumeration.

### Reporting summary

Further information on research design is available in the Nature Research Reporting Summary linked to this article.

### Data availability

The data supporting the findings of this paper are available within the Article and its Supplementary Information, which contains raw images of gels. The atomic coordinates have been deposited at the PDB (7THM). The 3D cryo-EM map has been deposited in the Electron Microscopy Data Bank under the accession number EMD-25898. Source data are provided with this paper.

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

**Acknowledgements** We thank the members of the Tagliabracci laboratory for discussions; D. Karlin for notifying us of the similarity between SELO and the NiRAN domain; B. Park for help with kinetics; A. Lemoff for help with intact-mass analysis; J. Kilgore, N. Williams and the staff at the UTSW Preclinical Pharmacology Core for detection and quantification of GpppA; S. Wilson and S. Rihn for the SARS-CoV-2 infectious clone and for technical guidance; and the staff at the Structural Biology Laboratory and the Cryo-Electron Microscopy Facility at UT Southwestern Medical Center (partially supported by grant RP170644 from the Cancer Prevention & Research Institute of Texas (CPRIT)) for cryo-EM studies. A portion of this research was supported by the W. M. Keck Foundation Medical Research Grant (to V.S.T., K.P. and J.W.S.), the National Institutes of Health grant R01GM135189 (to V.S.T.), 1DP1AI158124 (to J.W.S.), a Welch Foundation Grant I-1911 (to V.S.T.), a Life Sciences Research Foundation Fellowship (to G.H.) and a Polish National Agency for Scientific Exchange scholarship PPN/BEK/2018/1/00431 (to K.P.). This study made use of the National Magnetic Resonance Facility at Madison, which is supported by NIH grant R24GM141526. J.W.S. is a Burroughs Wellcome Fund Investigator in the Pathogenesis of Infectious Disease. V.S.T. is a Howard Hughes Medical Institute Investigator, a Michael L. Rosenberg Scholar in Medical Research, a CPRIT Scholar (RR150033) and a Searle Scholar.

**Author contributions** G.J.P., A.O., G.H., J.W.S. and V.S.T. designed the experiments. G.J.P., A.O., G.H., J.L.E., A.M. and V.S.T. conducted experiments. G.J.P. discovered the RNAylation and GDP-PRNTase activity of the NiRAN domain. A.O., Z.C. and Y.L. performed the cryo-EM. A.O. performed methylation experiments. G.H. performed GTase and nsp9 NMPylation experiments. A.M. performed intact-mass data analysis. M.T. and K.H.-W. performed NMR experiments. K.P. performed the bioinformatics. G.J.P., A.O. and G.H. purified proteins. G.J.P., A.O. and V.S.T. performed cloning and site-directed mutagenesis. J.L.E. performed infectious SARS-CoV-2 experiments. G.J.P., A.O., K.P., J.W.S. and V.S.T. wrote the manuscript with input from all of the authors.

**Competing interests** The authors declare no competing interests.

**Additional information**
**Correspondence and requests for materials** should be addressed to Vincent S. Tagliabracci.

**Extended Data Fig. 1 | Sequence alignment of the NiRAN domain reveals similarity to the pseudokinase selenoprotein O (SELO).** Multiple sequence alignment highlighting conserved kinase-like active site residues in the NiRAN domain among several CoVs, other selected *Nidovirales* (Arteri-, Mesoni- and Roniviruses) and the human SELO pseudokinase. Top: amino acid sequence surrounding the Lys-Glu ion pair. Bottom: and the amino acid sequence surrounding the active site and the "DFG" motif.

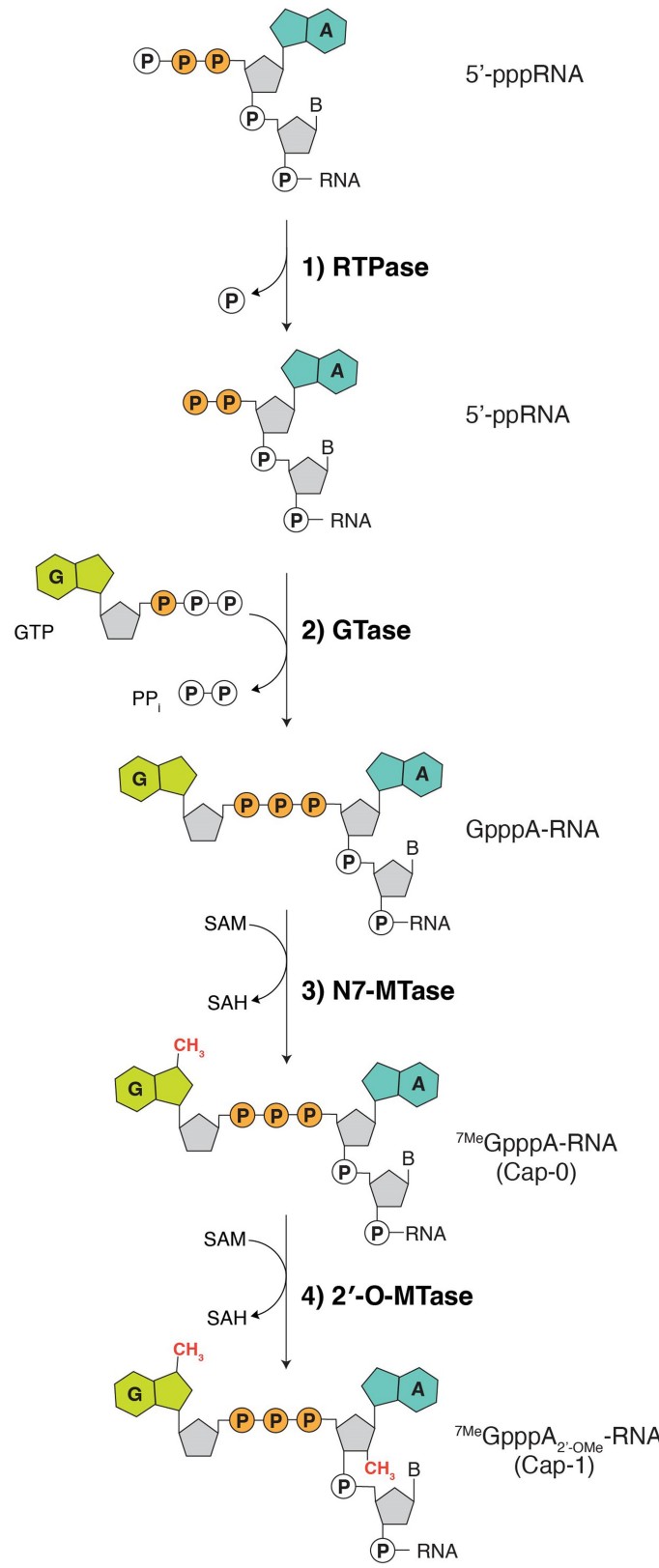

5'-pppRNA

**1) RTPase**

5'-ppRNA

GTP

**2) GTase**

PP_i

GpppA-RNA

SAM

SAH

**3) N7-MTase**

CH_3

$^{7Me}$GpppA-RNA
(Cap-0)

SAM

SAH

**4) 2′-O-MTase**

CH_3

CH_3

$^{7Me}$GpppA$_{2'-OMe}$-RNA
(Cap-1)

**Extended Data Fig. 2** | See next page for caption.

**Extended Data Fig. 2 | The canonical eukaryotic mRNA capping mechanism.** The $^{7Me}$GpppA$_{2'-O-Me}$ cap on eukaryotic RNA is formed co-transcriptionally by four enzymes: **1)** an RNA triphosphatase (RTPase), which removes the γ-phosphate from the nascent 5′-triphosphorylated RNA (5′-pppRNA) to yield a 5′-diphosphorylated RNA (5′-ppRNA); **2)** a guanylyltransferase (GTase), which transfers the GMP moiety from GTP to the 5′-ppRNA to form the core cap structure GpppN-RNA; **3)** a (guanine-N7)-methyltransferase (N7-MTase), which methylates the cap guanine at the N7 position; and **4)** a (nucleoside-2′-O)-methyltransferase (2′-O-MTase), which methylates the ribose-2′-OH position on the first nucleotide of the RNA. B denotes any base; GTP, Guanosine triphosphate; GDP, Guanosine diphosphate; PP$_i$, pyrophosphate; SAM, S-Adenosyl methionine.

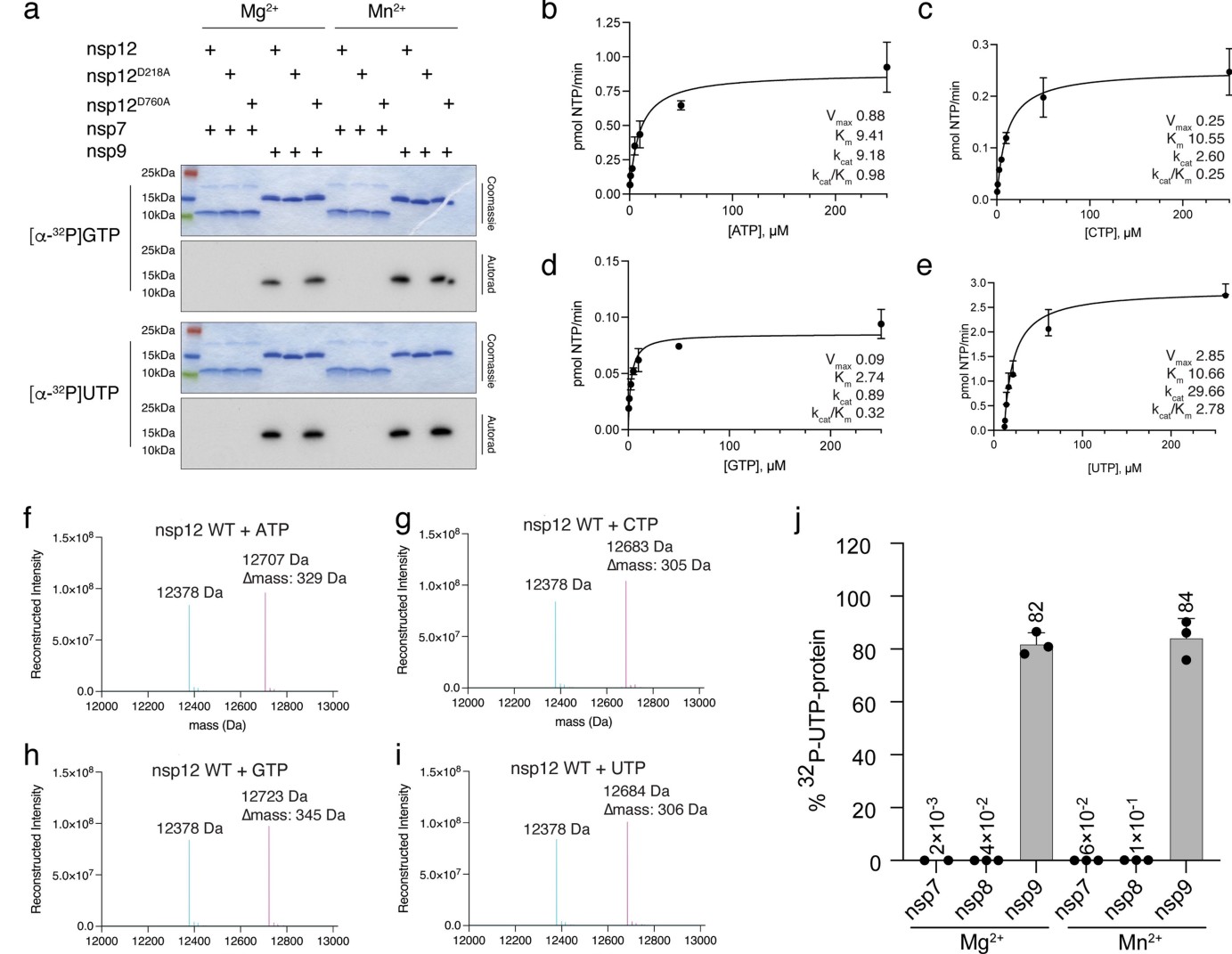

**Extended Data Fig. 3 | The NiRAN domain NMPylates nsp9. a.** Incorporation of α-32P from [α-32P]GTP or [α-32P]UTP into nsp7 or nsp9 by WT nsp12, the NiRAN mutant (D218A), or the polymerase mutant (D760A). Reactions were performed in the presence of $Mg^{2+}$ or $Mn^{2+}$ and the products were resolved by SDS-PAGE and visualized by Coomassie staining (top) and autoradiography (bottom). **b–e.** Kinetic analysis depicting the concentration dependence of (**b**) ATP, (**c**) CTP, (**d**) GTP, or (**e**) UTP on the rate of nsp9 NMPylation by the NiRAN domain. $K_m$ (μM), $V_{max}$ (pmol NTP/min), $k_{cat}$ (min−1) and $k_{cat}/K_m$ (min−1/μM) values are indicated. Plots shown are the mean and SD of triplicate reactions. **f–i.** Intact mass LC/MS spectra of unmodified nsp9 (*cyan*) overlayed with NMPylated nsp9

(*pink*) following incubation with WT nsp12 and (**f**) ATP, (**g**) CTP, (**h**) GTP, or (**i**) UTP. The observed masses are shown in the insets. The theoretical mass of unmodified nsp9 is 12378.2 Da and the theoretical increase in mass with the addition of each NMP is as follows: AMP, 329 Da; CMP, 305 Da; GMP, 345 Da; UMP, 306 Da. **j.** Quantification of the stoichiometry of NMPylation of nsp7, nsp8 and nsp9. Reactions were performed in the presence of $Mg^{2+}$ or $Mn^{2+}$. Reaction products were resolved by SDS-PAGE, the corresponding protein bands were excised from the gel, and radioactivity was quantified by scintillation counting. Bars: mean ± SD of three independent experiments.

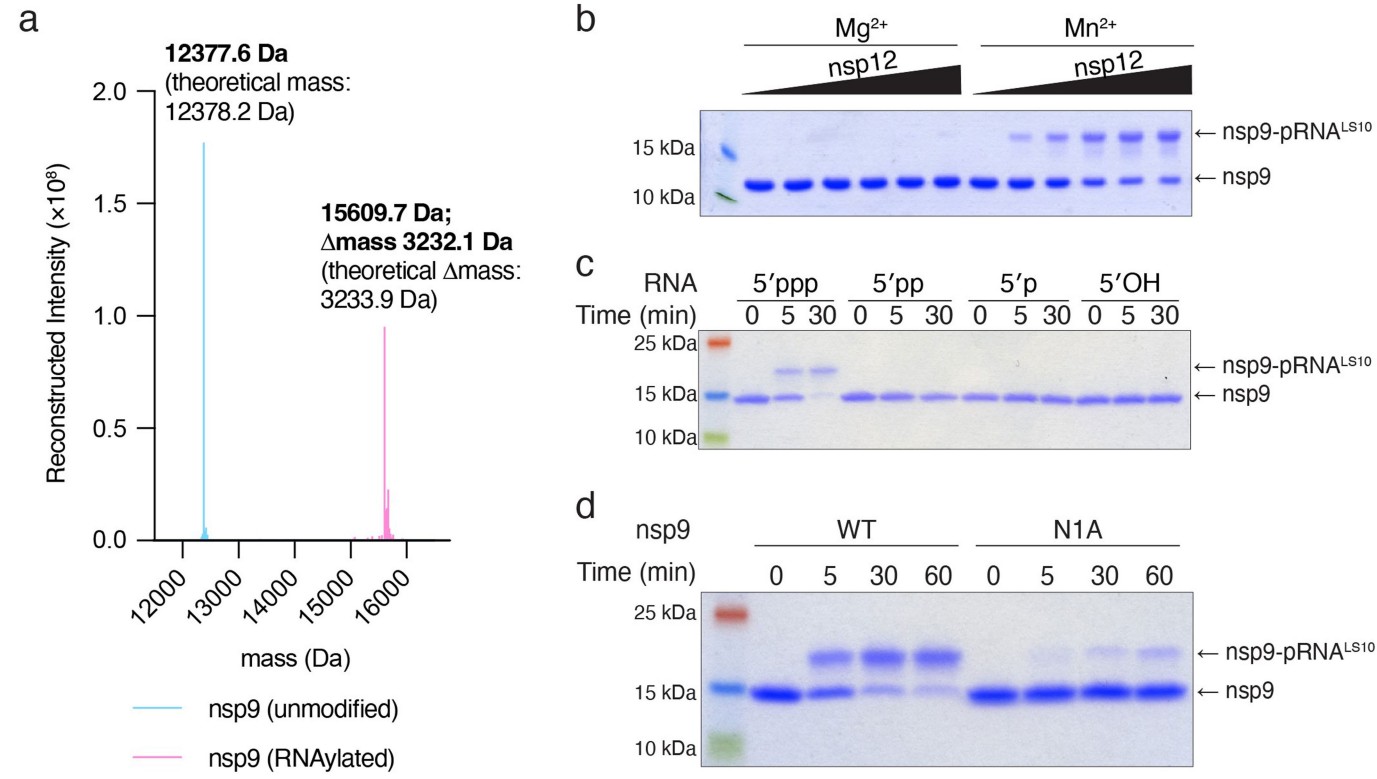

**Extended Data Fig. 4 | Characterization of NiRAN RNAylation activity.**
**a**. Intact mass LC/MS spectra (overlayed) of unmodified nsp9 (*cyan*) or nsp9 after incubation with 5′-pppRNA^LS10 and WT nsp12 (*pink*). The theoretical and observed masses are shown in the insets. The Δmass of 3233.17 Da corresponds to monophosphorylated RNA^LS10 (5′-pRNA^LS10). **b**. Incorporation of RNA into nsp9 by nsp12 (0–4 μM) in the presence of Mg²⁺ or Mn²⁺. Reaction products were analysed as in Fig. 2b. **c**. Incorporation of RNA with the indicated 5′ ends into nsp9 by nsp12. Reaction products were analysed as in Fig. 2b. **d**. Time-dependent incorporation of 5′-pRNA^LS10 into nsp9 or the nsp9 N1A mutant. Reaction products were analysed as in Fig. 2b.

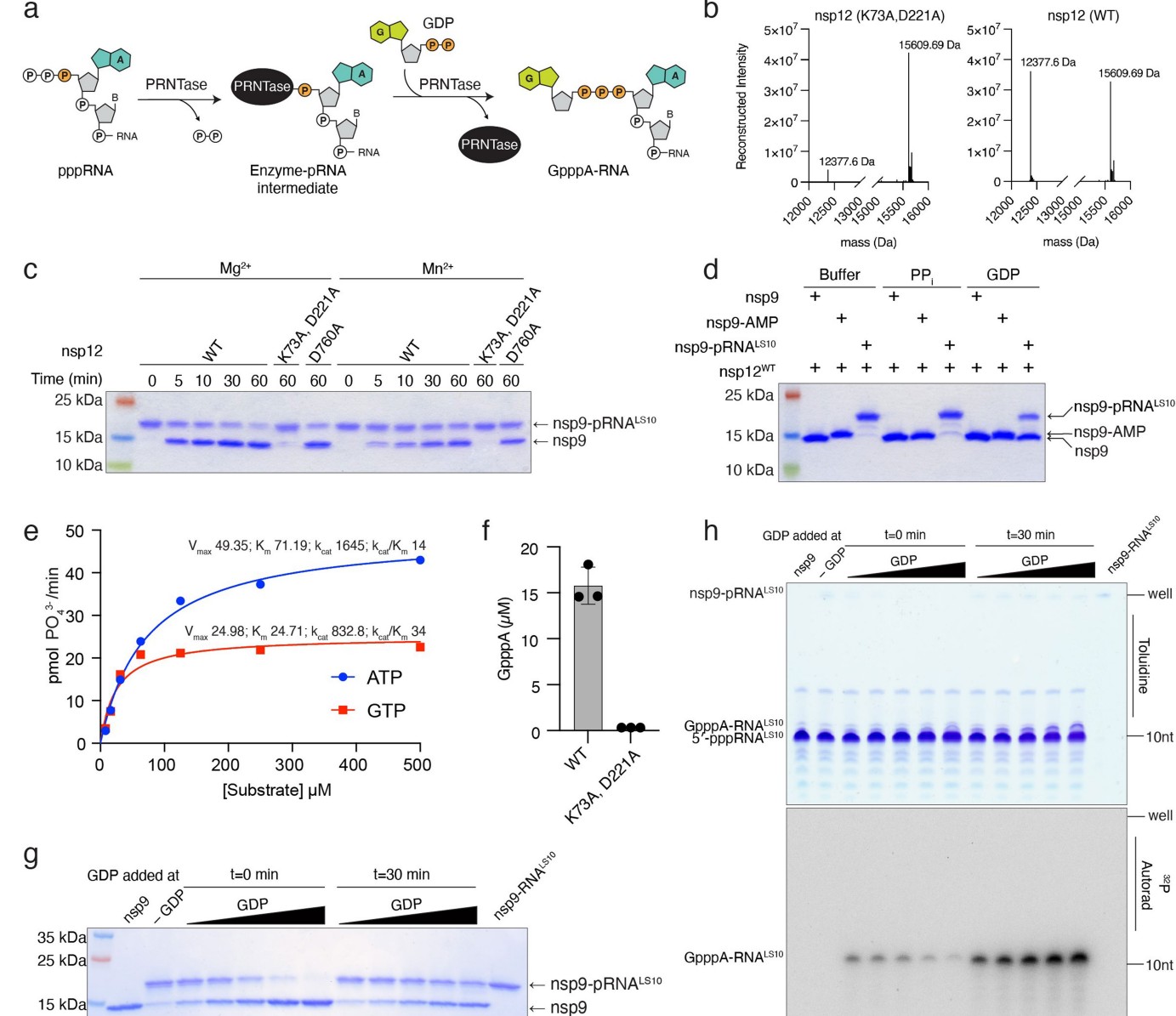

**Extended Data Fig. 5 | Characterization of nsp12 NiRAN GDP-PRNTase activity. a**. Schematic representation depicting the mechanism of core cap formation by vesicular stomatitis virus (VSV) polyribonucleotidyltransferase (PRNTase). **b**. Intact mass LC/MS spectra of nsp9–pRNA$^{LS10}$ after incubation with GDP and WT nsp12 (*right*) or the NiRAN mutant (*left*). The theoretical mass of nsp9 is 12378.2 Da and the theoretical mass of nsp9–pRNA$^{LS10}$ is 15611.5 Da. **c**. Time-dependent deRNAylation of nsp9–pRNA$^{LS10}$ by WT nsp12, the NiRAN mutant (K73A, D221A), or the polymerase mutant (D760A) in the presence of GDP and either Mg$^{2+}$ or Mn$^{2+}$. Reaction products were analysed as in Fig. 2b. **d**. NiRAN-dependent deAMPylation or deRNAylation of nsp9 in the presence of PP$_i$ or GDP. Reaction products were analysed as in Fig. 2b. **e**. Concentration dependence of ATP (*blue*) or GTP (*red*) on the rate of phosphate release

catalysed by nsp13. V$_{max}$ (pmol PO$_4^{3-}$/min), K$_m$ (µM), k$_{cat}$ (min$^{-1}$) and k$_{cat}$/K$_m$ (min$^{-1}$/µM) values are indicated. **f**. HPLC/MS quantification of GpppA formed during the NiRAN-catalysed deRNAylation of nsp9–pRNA$^{LS10}$. Reaction products were digested with nuclease P1 prior to HPLC analysis. Reactions were performed in triplicate and error bars represent the standard deviation. **g**, **h**. NiRAN-catalysed capping reactions depicting the inhibitory effect of GDP on RNAylation. nsp9 was incubated with excess 5′-pppRNA$^{LS10}$ in presence of nsp12 with no GDP (-GDP), or increasing concentrations (6.25-100 µM) of [α-$^{32}$P]GDP added either at time zero (t = 0 min), or after the RNAylation reaction was allowed to proceed for 30 min (t=30 min). Reaction products were analysed by SDS-PAGE **(g)**, and urea–PAGE **(h)**.

a

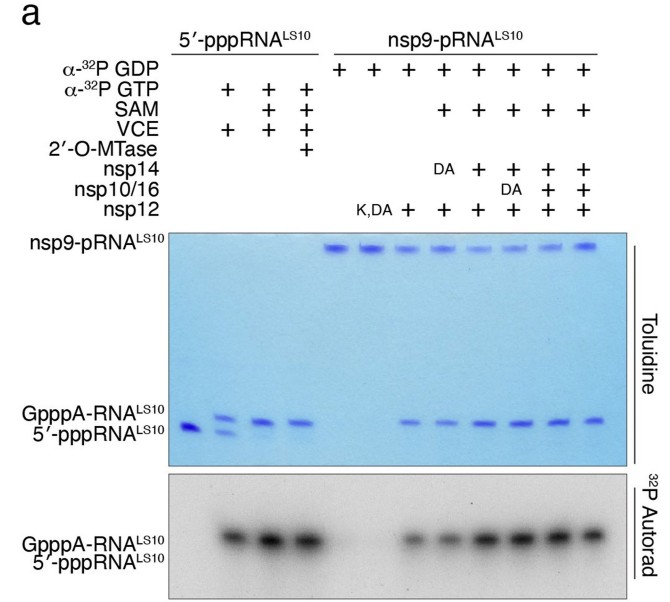

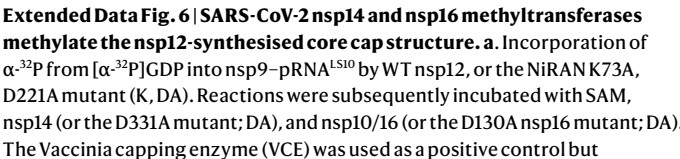

b

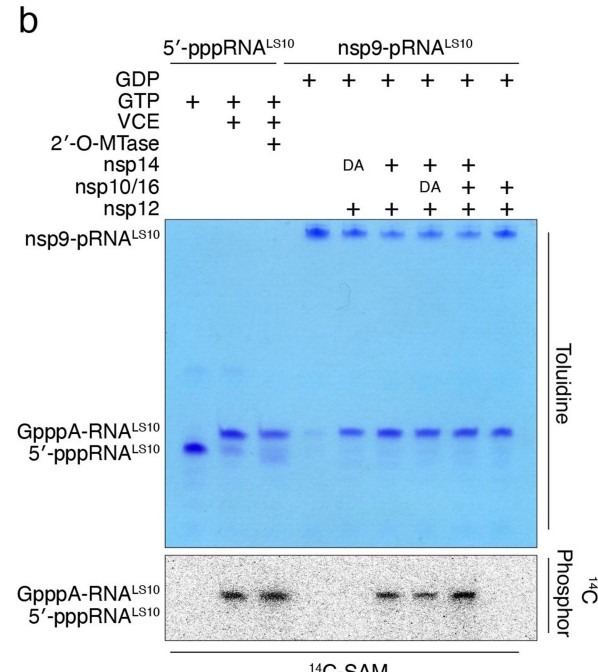

**Extended Data Fig. 6 | SARS-CoV-2 nsp14 and nsp16 methyltransferases methylate the nsp12-synthesised core cap structure. a.** Incorporation of α-$^{32}$P from [α-$^{32}$P]GDP into nsp9–pRNA$^{LS10}$ by WT nsp12, or the NiRAN K73A, D221A mutant (K, DA). Reactions were subsequently incubated with SAM, nsp14 (or the D331A mutant; DA), and nsp10/16 (or the D130A nsp16 mutant; DA). The Vaccinia capping enzyme (VCE) was used as a positive control but incubated with [α-$^{32}$P]GTP and the Vaccinia 2′-O-MTase. Reaction products were analysed as in Fig. 3d. **b.** Incorporation of $^{14}$C from [methyl-$^{14}$C]SAM into GpppA-RNA$^{LS10}$ by nsp14 (WT or the D331A mutant; DA), and nsp10/16 (WT or the D130A nsp16 mutant; DA). The VCE and the Vaccinia 2′-O-MTase were used as positive controls. Reaction products were analysed as in Fig. 3d. The $^{14}$C signal was detected by phophorimaging.

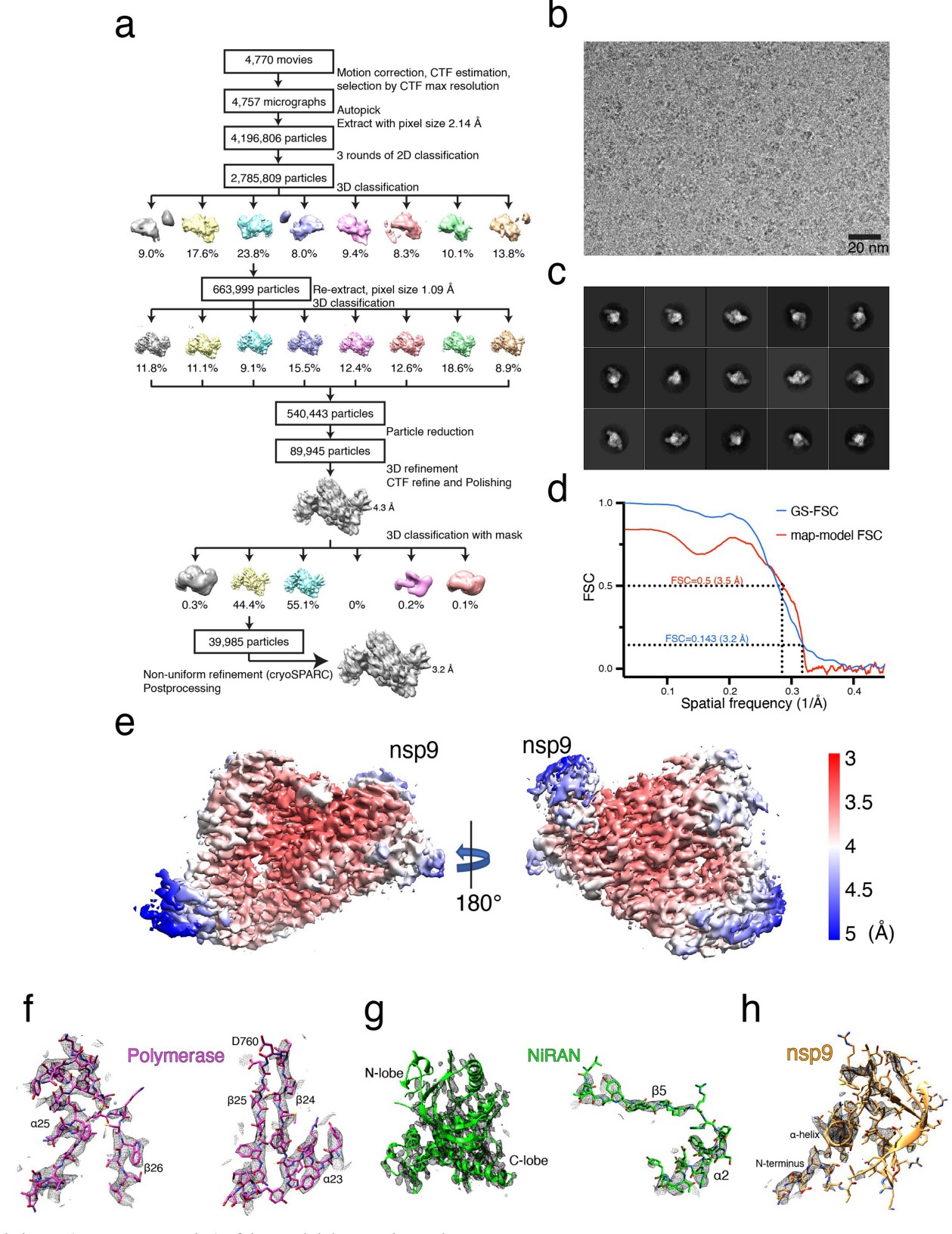

**Extended Data Fig. 7 | Cryo-EM analysis of the nsp7/8/9/12 complex. a.** Flow chart representing data processing for the nsp7/8/9/12 complex. **b.** A representative micrograph of the nsp7/8/9/12 complex grids. **c.** Representative 2D classes generated by RELION 2D-classification. **d.** Gold-standard FSC curve (blue), and map-model FSC curve (red). Curves were generated by cryoSPARC and Phenix suite, respectively. **e.** Local resolution of the nsp7/8/9/12 complex calculated by RELION from final cryoSPARC half-maps. Position of nsp9 is indicated. **f-h.** Exemplary cryo-EM density (black mesh) on (**f**) Polymerase (Magenta), (**g**) NiRAN (green) and (**h**) nsp9 (gold). Note weaker density in the N-lobe of the kinase-like NiRAN domain (left panel of **g**, compare top to bottom of the image), and poor density in nsp9 (**h**), in areas not in direct contact with the NiRAN domain.

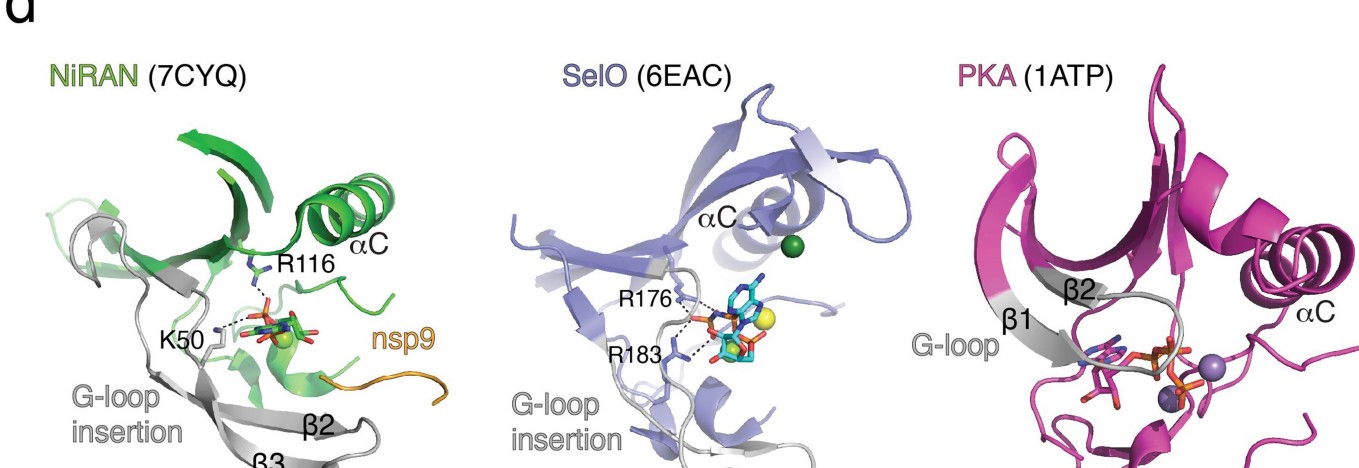

**Extended Data Fig. 8 | Biochemical analysis of the isolated NiRAN domain and comparison of the kinase-like domains of nsp12 and SELO. a**. Size exclusion chromatography analysis of the isolated NiRAN domain (residues 1–326; ΔRdRp). Standards are shown. **b**. Incorporation of 5′-pRNA$^{LS10}$ into nsp9 and deRNAylation of nsp9–pRNA$^{LS10}$ by WT nsp12, the NiRAN mutant (D218A), the polymerase mutant (D760A), or the ΔRdRP mutant. Reaction products were analysed as in Fig. 2b. **c**. Cartoon representation comparing the NiRAN active site catalytic residues (green) to the active site residues in SELO (purple). The divalent cations are shown as spheres. **d**. Comparison of the Gly-rich loop regions in NiRAN (PDB: 7CYQ; left), SELO (PDB: 6EAC), and the canonical kinase PKA (PDB: 1ATP; right). Green sphere – Mg$^{2+}$, dark green sphere – Chloride, yellow sphere – calcium, violet sphere – Mn$^{2+}$.

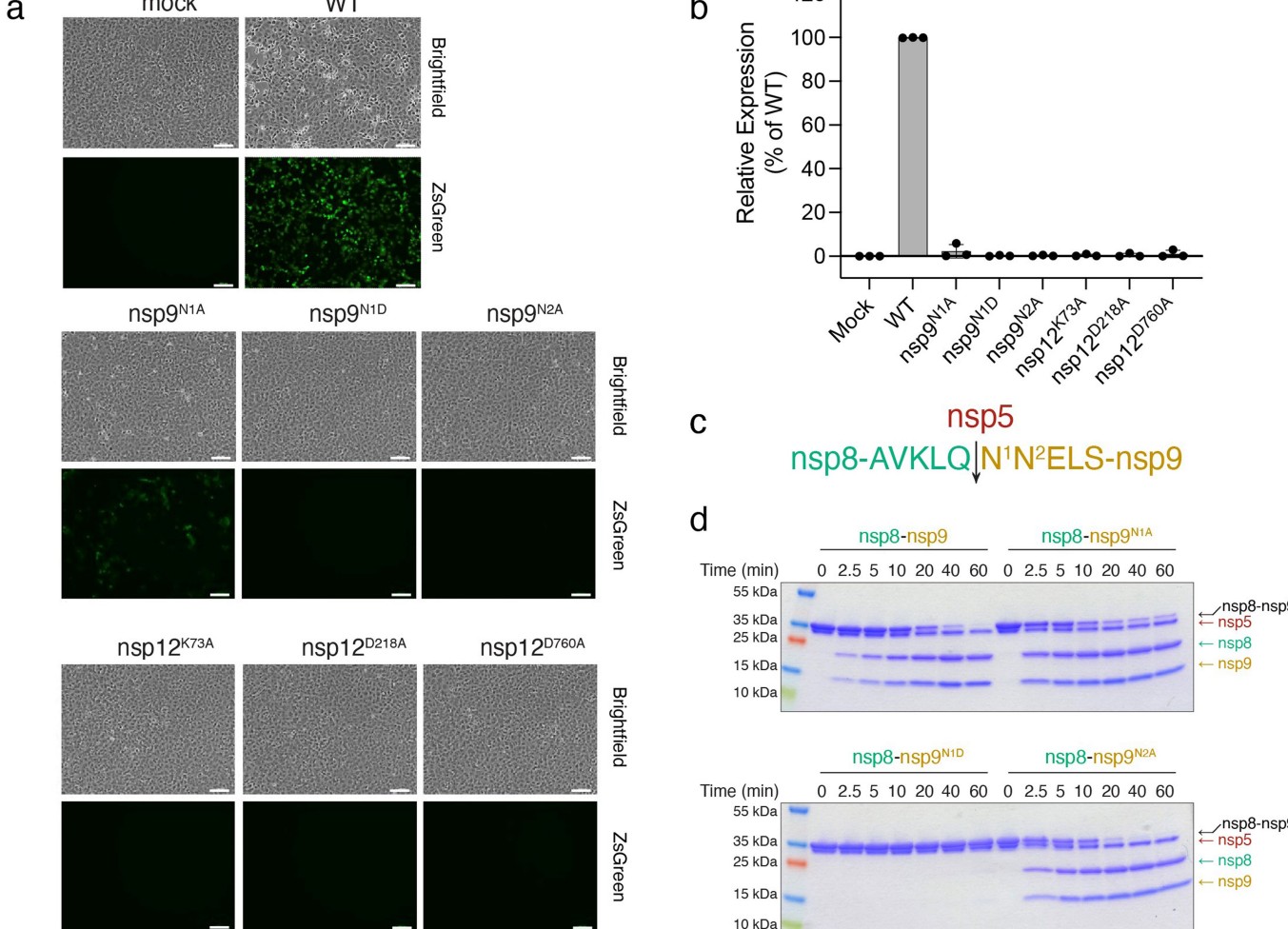

**Extended Data Fig. 9 | Genetic insights into RNA capping by the NiRAN domain. a**. Microscopy images showing brightfield (upper) or fluorescence-based images (ZsGreen; lower) of SARS-CoV-2-ZsGreen production in VeroE6-C1008-TMPRSS2 cells. Mock-transfected panels were incubated with transfection reagents lacking DNA. The mutations engineered into either nsp9 or nsp12 are indicated above each set of images. Data are one set of images representative of three independent biological replicates. **b**. Relative viral yields from WT or mutant SARS-CoV-2 viruses bearing indicated mutations in nsp9 and nsp12. Viral RNA levels were quantified by RT-qPCR. Bars: mean ± SD of N = 3 biological replicates. **c**. The amino acid sequence between nsp8 (green) and nsp9 (gold) depicting the cleavage site for the nsp5 (dark red) protease. N1 and N2 of nsp9 are highlighted. The arrow denotes the location of cleavage. **d**. Time-dependent proteolysis of the nsp8-nsp9 fusion protein by nsp5. Reaction products were separated by SDS PAGE and visualized by Coomassie staining.

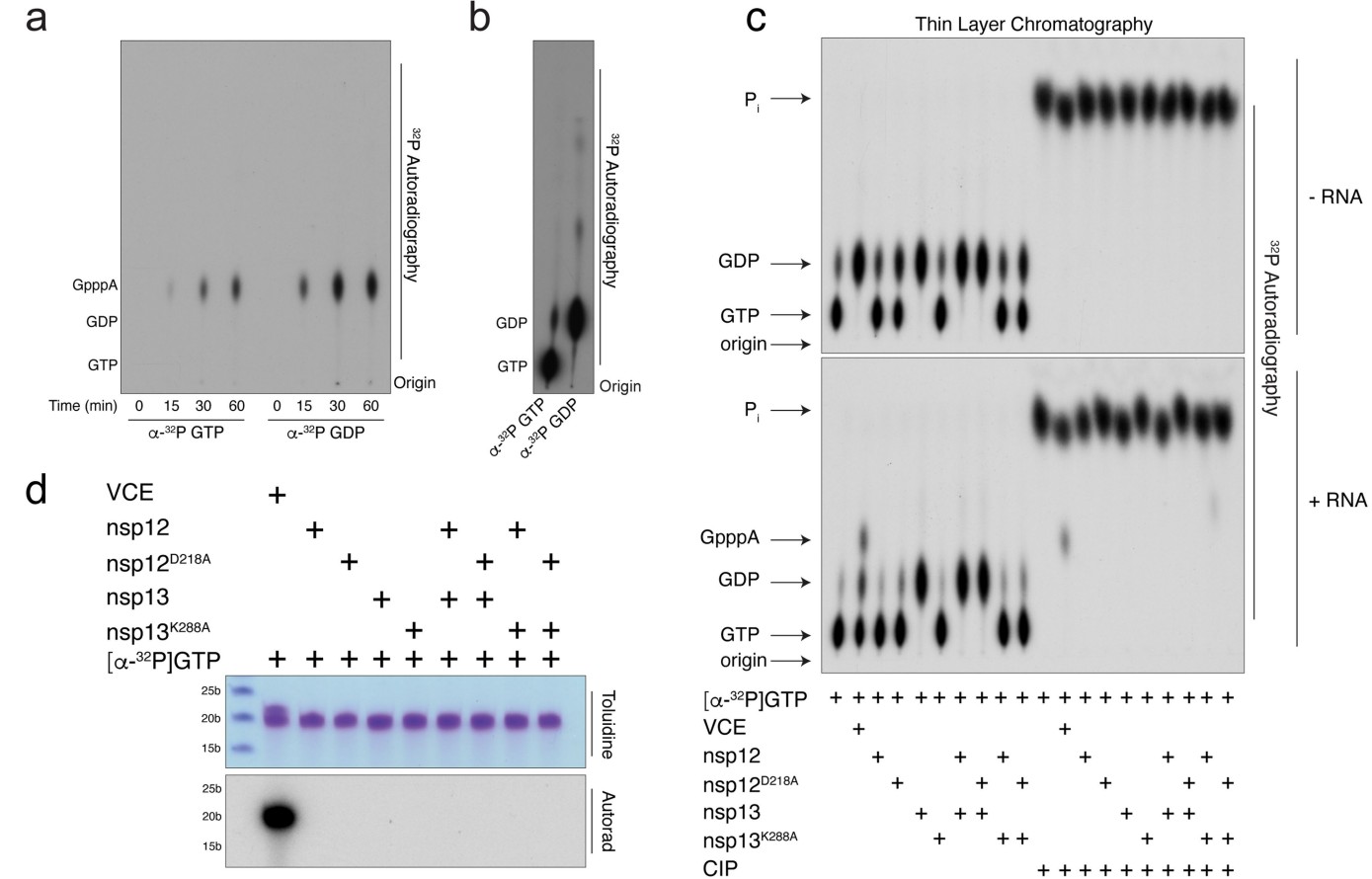

**Extended Data Fig. 10 | The NiRAN domain is specific for GDP in the capping reaction and does not transfer GMP from GTP to 5′-ppRNA. a**. Thin-layer chromatogram depicting the reaction products following incubation of nsp12 and nsp9–pRNA[LS10] with [α-32P]GTP or [α-32P]GDP for the indicated timepoints. Location of the cold standards (left) was visualized by UV fluorescence and the 32P by autoradiography. **b**. Thin-layer chromatogram depicting [α-32P]GTP or [α-32P]GDP used in the assays in **a**. Note the presence of GDP in the GTP sample. **c**. Thin-layer chromatograms depicting the reaction products resulting from the incubation of [α-32P]GTP with nsp12 or the inactive NiRAN mutant (D218A).

Reactions were performed as described in[26] with (*lower*) or without (*upper*) 5′-pppRNA[A19C] and included nsp13 or the inactive mutant (K288A) as indicated. Vaccinia capping enzyme (VCE) was used as a positive control. Reaction products were digested with nuclease P1, then treated with or without calf intestinal alkaline phosphatase (CIP) and analysed by PEI-cellulose thin-layer chromatography followed by autoradiography. The positions of the origin and standard marker compounds are indicated. **d**. RNA products from **c** were analysed by TBE urea–PAGE and visualized by toluidine blue O staining (*upper*) and autoradiography (*lower*). Markers indicate RNA size by base length.

# Reporting Summary

## Statistics

For all statistical analyses, confirm that the following items are present in the figure legend, table legend, main text, or Methods section.

| n/a | Confirmed | |
|---|---|---|
| ☐ | ☒ | The exact sample size (*n*) for each experimental group/condition, given as a discrete number and unit of measurement |
| ☐ | ☒ | A statement on whether measurements were taken from distinct samples or whether the same sample was measured repeatedly |
| ☒ | ☐ | The statistical test(s) used AND whether they are one- or two-sided<br>*Only common tests should be described solely by name; describe more complex techniques in the Methods section.* |
| ☒ | ☐ | A description of all covariates tested |
| ☒ | ☐ | A description of any assumptions or corrections, such as tests of normality and adjustment for multiple comparisons |
| ☐ | ☒ | A full description of the statistical parameters including central tendency (e.g. means) or other basic estimates (e.g. regression coefficient) AND variation (e.g. standard deviation) or associated estimates of uncertainty (e.g. confidence intervals) |
| ☒ | ☐ | For null hypothesis testing, the test statistic (e.g. *F*, *t*, *r*) with confidence intervals, effect sizes, degrees of freedom and *P* value noted<br>*Give P values as exact values whenever suitable.* |
| ☒ | ☐ | For Bayesian analysis, information on the choice of priors and Markov chain Monte Carlo settings |
| ☒ | ☐ | For hierarchical and complex designs, identification of the appropriate level for tests and full reporting of outcomes |
| ☒ | ☐ | Estimates of effect sizes (e.g. Cohen's *d*, Pearson's *r*), indicating how they were calculated |

*Our web collection on statistics for biologists contains articles on many of the points above.*

## Software and code

Policy information about availability of computer code

| Data collection | 1) Cryo-EM: SerialEM 4.0.4 (ref 46). |
|---|---|
| Data analysis | 1) NMR: All 2D spectra were processed using NMRPipe 11.0 (ref. 42) and analysed with NMRFAM-SPARKY 1.3 (ref 43).<br>2) Intact mass spectrometry: The acquired mass spectra for the proteins of interest were deconvoluted using BioPharmaView v. 3.0.1 software (Sciex) in order to obtain the molecular weights.<br>3) HPLC/MS: Peak areas were determined and data were further analysed using the Sciex Analyst 1.7.2 software package.<br>4) GraphPad Software 9.4.0 , San Diego, California USA, www.graphpad.com<br>5) Cryo-EM: Relion 3.1 (ref. 47); MotionCor2 1.5 (ref. 48); Gctf 1.06 (ref. 49); crYOLO 1.7 (ref. 50); cryoSPARC 3.2 (ref. 51); Coot 0.9.6 (ref. 52); Phenix 1.20.1-4487 (ref. 53); MolProbity 4.5.1 (ref. 54).<br>6) Bioinformatics: FATCAT 2.0 (ref. 55); ESPript 3 (ref. 56)<br>7) Fiji 2.1.0/1.53h (ref. 44) |

For manuscripts utilizing custom algorithms or software that are central to the research but not yet described in published literature, software must be made available to editors and reviewers. We strongly encourage code deposition in a community repository (e.g. GitHub). See the Nature Portfolio guidelines for submitting code & software for further information.

## Data

Policy information about availability of data

 All manuscripts must include a data availability statement. This statement should provide the following information, where applicable:
  - Accession codes, unique identifiers, or web links for publicly available datasets
  - A description of any restrictions on data availability
  - For clinical datasets or third party data, please ensure that the statement adheres to our policy

Cryo-EM model has been deposited in the Protein Data Bank under accession number 7THM. Cryo-EM density maps have been deposited in the Electron Microscopy Data Bank with accession codes EMD-25898. All other data generated during and/or analysed during the current study are available from the corresponding author on reasonable request.
Additionally, publicly available datasets were used:
Pseudomonas syringae SelO protein crystal structure PDBID: 6EAC
Mouse PKA C-alpha crystal structure PDBID: 1ATP
Extended SARS-CoV-2 RTC complex structure with non-native nsp9 PDBID: 7CYQ, EMDB: EMD-30504

# Field-specific reporting

Please select the one below that is the best fit for your research. If you are not sure, read the appropriate sections before making your selection.

☒ Life sciences    ☐ Behavioural & social sciences    ☐ Ecological, evolutionary & environmental sciences

For a reference copy of the document with all sections, see nature.com/documents/nr-reporting-summary-flat.pdf

# Life sciences study design

All studies must disclose on these points even when the disclosure is negative.

| | |
|---|---|
| Sample size | No sample size was predetermined. Sample sizes were chosen according to the standard generally accepted in the field. |
| Data exclusions | No data were excluded from the analyses. |
| Replication | Each experiment presented in the paper was repeated at least twice, with similar results. Reported results were consistently replicated across all experiments. |
| Randomization | No randomization was used in performing in vitro biochemical experiments. Covariate control is irrelevant for this study. No human or animal subjects were used in the study. |
| Blinding | Blinding was not used in this study. Due to the experimental setup of the in-vitro experiments, blinding was not possible, or would not provide benefit. |

# Reporting for specific materials, systems and methods

We require information from authors about some types of materials, experimental systems and methods used in many studies. Here, indicate whether each material, system or method listed is relevant to your study. If you are not sure if a list item applies to your research, read the appropriate section before selecting a response.

### Materials & experimental systems

| n/a | Involved in the study |
|---|---|
| ☒ | ☐ Antibodies |
| ☐ | ☒ Eukaryotic cell lines |
| ☒ | ☐ Palaeontology and archaeology |
| ☒ | ☐ Animals and other organisms |
| ☒ | ☐ Human research participants |
| ☒ | ☐ Clinical data |
| ☒ | ☐ Dual use research of concern |

### Methods

| n/a | Involved in the study |
|---|---|
| ☒ | ☐ ChIP-seq |
| ☒ | ☐ Flow cytometry |
| ☒ | ☐ MRI-based neuroimaging |

## Eukaryotic cell lines

Policy information about cell lines

| | |
|---|---|
| Cell line source(s) | BHK-21J cells were a generous gift from C. Rice, not commercially sourced. VeroE6-C1008 cells were obtained from ATCC. |

Authentication

These are not human cells and thus we do not do STR profiling. The ATCC authenticates VeroE6-C1008. BHK-21J were not authenticated.

Mycoplasma contamination

All cell lines are routinely tested for mycoplasma using a sensitive PCR-based assay. The cells in this study tested negative for mycoplasma.

Commonly misidentified lines
(See ICLAC register)

No commonly misidentified lines were used.

