## [Peer Review File · Nature]

Manuscript Title: The mechanism of RNA capping by SARS-CoV-2

Reviewer Comments & Author Rebuttals

Reviewer Reports on the Initial Version:

Referees' comments:

Referee #1 (Remarks to the Author):

This piece of research provides evidence for how SARS-Cov-2 catalyses the RNA capping reaction, to create guanosine triphosphate linkage to the first transcribed nucleotide. How coronaviruses catalyse cap addition has been unclear and the subject of much speculation. This research is novel and important given covid-19 pandemic.

The biochemistry is thorough and convincing. The data is clear. The paper is well written. Structural biology is not my area so I can say that it appears appropriate and informative.

Overall I think this is an important piece of work for the research community to receive promptly

Referee #2 (Remarks to the Author):

This is a beautiful biochemical and structural study of the replication and transcriptional machinery of SARS-CoV-2 that solves the long standing puzzle of the mechanism by which the messenger RNA caps present at the 5' end of the viral mRNA are formed. In eukaryotic cells and for most viruses, mRNA cap formation proceeds through the sequential action of a set of enzymes. An RNA triphosphatase trims the terminal phosphate from a 5' pppRNA to generate 5' ppRNA which is then capped by an RNA guanylyltransferase that transfers GMP derived from GTP to generate a 5'-5' GpppRNA cap structure via a covalent adduct that is formed between the GMP and a catalytic lysine residue. A guanine-N7 methylase modifies the cap to transfer a methyl group from AdoMet generate 7mGpppRNA and AdoHcy which is subsequently modified by a usually separate ribose 2'-O methylase that transfers a second methyl group from AdoMet to generate 7mGpppNmN. Coronaviruses generate the same structure on their mRNAs but the identification of an RNA guanylyltransferase has proved elusive. In this study, the authors provide compelling evidence for an unconventional mechanism of messenger RNA cap formation. They provide detailed biochemical and structural evidence that the N-terminus of the nsp12 polymerase that contains a poorly understood domain termed the NiRAN (Nidovirus RDRP associated nucleotidyltransferase) catalyzes the formation of a nsp9-pRNA intermediate in mRNA cap formation in which the RNA forms an adduct with the N terminal amino acid residue of nsp9, and addition of GDP results in transfer of the monophosphate RNA onto GDP to form the mRNA cap-structure. This mechanism is reminiscent but distinct from that used by vesicular stomatitis and rabies viruses – and presumably other nonsegmented negative-strand RNA viruses – to cap their mRNAs whereby the mRNA capping domain of their polymerase proteins forms a covalent adduct with the nascent mRNA strand

through an active site histidine residue and transfers the monophosphate RNA onto a GDP acceptor. A further similarity with that mechanism pertains to the sequence specific nature of the capping reaction – in that substitution of the first or second nucleotides in the pppRNA substrate interferes with the reaction. The present study is very well carried out, likely to be of broad interest and will undoubtedly stimulate a robust effort to target the novel enzymatic activities involved in cap formation in the quest for new antivirals.

Although the study is thorough and well conducted, there are some points that would benefit from being addressed.

1: Previous work has implicated nsp13 as the RNA triphosphatase. The presence of an RNA triphosphatase supports a conventional strategy of mRNA cap formation involving the RTPase and GTase. Here the authors posit that the function of nsp13 is to provide the GDP derived from GTP for the capping reaction. Although they present a reasoned argument to support this model, no experimental tests are carried out. A direct molecular test of this would further strengthen the authors claims. At present although the authors show that provision of radioactive GDP to the covalent nsp9-pRNA resolves to generate an RNA product that is consistent with formation of a capped RNA, they do not show that nsp13 is the source of GDP.

2: To support the importance of the predicted catalytic residues in cap formation the authors engineer an infectious molecular clone of SARS-CoV-2 to mutate the N-terminal residue of nsp9 N1 to A or D, or N2 to A, and also mutate the catalytic residues K73A and D218A in the NiRAN domain, and D760A of the RdRP domain of nsp12. They assess the effect of those mutations on viral replication using a RT-PCR assay and show that the levels of RNA are reduced 400-4000 fold. The choice of assay here is confusing. One would anticipate that failure to cap the RNA would be lethal for viral replication, rather than resulting in a 2-3 log reduction in viral RNA levels. The viral infectivity should be reported. As viral RNA is produced from the various mutants it would be of significant interest to know whether the RNAs are actually capped or whether they are capped aberrantly. One would also anticipate that production of uncapped RNA would potentially activate antiviral signaling which could readily be measured in the context of the mutants.

3: Inclusion of quantitative measurements in the main figures would aid the reader – for example, in Figure 1 what is the level of complex formation with nsp9 and the various mutants? Similarly in Figure 2 for the various mutations in the leader RNA sequence. Similarly Figure 5g and h.

4: Although only a minor point, the main text introduces mRNA cap formation (lines 77-87) in eukaryotic cells. The cap structure was first discovered on viral mRNAs (reovirus and vaccinia) and only subsequently were the details of how the process works in eukaryotes defined. The introduction would benefit from a brief description of this historical context.

Additional minor text issues included

line 37 “how this cap is made is not completely understood”

line 54 “topical of these”

line 61-62 “several promising antivirals used to treat COVID-19 including” Although remdesivir and molnupiravir are used clinically it is arguable as to whether they are promising.

Referee #3 (Remarks to the Author):

In the submitted manuscript Park and colleagues provide a detailed biochemical characterization of key steps involved in viral RNA capping. They show biochemical evidence that the NiRAN domain of Nsp12 harbors an unexpected RNAylation activity which is crucial for the formation of GpppA-RNA. The authors suggest the NiRAN domain first catalyzes the transfer of 5'-pRNA to the N-terminus of Nsp9. Subsequently, Nsp12 would facilitate the transfer of 5'-pRNA from Nsp9 to GDP, resulting in GpppA-RNA. Based on this product, the authors successfully recapitulate the formation of the cap-0 and cap-1 structure using recombinant viral proteins Nsp14 and Nsp16, respectively. However, the authors failed to generate a complete high-resolution structure of the E-RTC with native N-terminus of Nsp9. Thus, the authors rely on published data by Yan et al. 2021.

While the biochemical characterizations are well performed, and the manuscript is generally well written, its impact on the scientific community remains questionable. The authors do provide novel insights in the function of the NiRAN domain (RNAylation activity) and the involvement of Nsp9 in RNA capping. However, other aspects of the manuscript are confirmatory. I recommend this manuscript is considered for transfer to a more specialized Nature journal.

Besides that, several points have to be addressed to increase the clarity.

Line 57: "The SARS-CoV-2 RNA genome contains two open reading frames which are translated by host ribosomes to form two large polyproteins." This sentence is misleading since it suggests that the genome of SARS-CoV-2 only contains two ORFs, which is obviously not the case.

Line 92: "Furthermore, we present a cryo-EM structure of the SARS-CoV-2 RTC with the native N-terminus of nsp9 bound in the NiRAN active site." It should be specified here that almost all structural insights are based on previously published data by Yan et al. 2021. The EM-density provided by this study differs only in the most N-terminal amino acids of Nsp9, which are actually crucial for the proposed mechanism. Unfortunately, the authors do state in line 195 that the local resolution of these residues is not sufficient to "distinguish its exact position". Thus, the EM-densities provided by this study offer limited new insights. Please clarify this in the text.

Line 100: "We observed NiRAN-dependent NMPylation of native nsp9, but not native nsp7 or nsp8" and the fact that formation of a GpppA cap was impossible without Nsp9, contradicts published results by Shannon, A. et al. 2021 and Yan, L. et al. 2021, respectively. Some aspects are indeed explainable by the proposed capping pathway of this study. However, it would be good to discuss these aspects in more detail.

Line 125 refers to different RNAylation efficiencies depending on the first nucleotide of the RNA substrate and suggests substrate-selectivity. This an interesting point that could be extended to the second nucleotide.

Throughout the manuscript the authors extensively test the influence of Mn^{2+} and Mg^{2+} on biochemical reactions. Accordingly, it would be interesting to better discuss the role of different divalent ions during the capping reactions.

Line 205 mentions a mutant version of Nsp12 that just contains the NiRAN domain (Δ RdRp; 1-326). This mutant does not show any signs of enzymatic activity, which is presented as conformation for the described interactions between Nsp12 and Nsp9. However, no other biochemical data which characterizes the Δ RdRp mutant is shown. So it remains unclear if the deletion of such a large domain results in a properly folded protein. This must at least be discussed.

Figure 3d suggests that deRNAylation of Nsp9 is specific for GDP but also to a significant extent for GTP. However, in all following reactions involving Nsp12 and Nsp9 the effect of GTP was not tested. An important claim of this manuscript is that Nsp13 is essential for the proposed capping reaction since it catalyzes the formation of GDP (an activity Nsp12 lacks). Thus it would be interesting to see if the NiRAN domain is indeed incapable of utilizing GTP as substrate for the formation of GpppA-RNA.

Figure 5j presents intriguing data which suggests that kinase-like residues of the NiRAN domain and the N-terminus of nsp9 are essential for SARS-CoV-2 replication. Unfortunately, relative viral yields of duplicates instead of triplicates are depicted. This results in very large error bars, making it impossible to differentiate between the individual effects. This must be clarified.

Author Rebuttals to Initial Comments:

Referees' comments:

Referee #1 (Remarks to the Author):

This piece of research provides evidence for how SARS-Cov-2 catalyses the RNA capping reaction, to create guanosine triphosphate linkage to the first transcribed nucleotide. How coronaviruses catalyse cap addition has been unclear and the subject of much speculation. This research is novel and important given covid-19 pandemic.

The biochemistry is thorough and convincing. The data is clear. The paper is well written. Structural biology is not my area so I can say that it appears appropriate and informative.

Overall I think this is an important piece of work for the research community to receive promptly

We thank the reviewer for the positive comments regarding our manuscript.

Referee #2 (Remarks to the Author):

This is a beautiful biochemical and structural study of the replication and transcriptional machinery of SARS-CoV-2 that solves the long standing puzzle of the mechanism by which the messenger RNA caps present at the 5' end of the viral mRNA are formed. In eukaryotic cells and for most viruses, mRNA cap formation proceeds through the sequential action of a set of enzymes. An RNA triphosphatase trims the terminal phosphate from a 5' pppRNA to generate 5' ppRNA which is then capped by an RNA guanylyltransferase that transfers GMP derived from GTP to generate a 5'-5' GpppRNA cap structure via a covalent adduct that is formed between the GMP and a catalytic lysine residue. A guanine-N7 methylase modifies the cap to transfer a methyl group from AdoMet generate 7mGpppRNA and AdoHcy which is subsequently modified by a usually separate ribose 2'-O methylase that transfers a second methyl group from AdoMet to generate 7mGpppNmN. Coronaviruses generate the same structure

on their mRNAs but the identification of an RNA guanylyltransferase has proved elusive. In this study, the authors provide compelling evidence for an unconventional mechanism of messenger RNA cap formation. They provide detailed biochemical and structural evidence that the N-terminus of the nsp12 polymerase that contains a poorly understood domain termed the NiRAN (Nidovirus RDRP associated nucleotidyltransferase) catalyzes the formation of a nsp9-pRNA intermediate in mRNA cap formation in which the RNA forms an adduct with the N terminal amino acid residue of nsp9, and addition of GDP results in transfer of the monophosphate RNA onto GDP to form the mRNA cap-structure. This mechanism is reminiscent but distinct from that used by vesicular stomatitis and rabies viruses – and presumably other nonsegmented negative-strand RNA viruses – to cap their mRNAs whereby the mRNA capping domain of their polymerase proteins forms a covalent adduct with the nascent mRNA strand through an active site histidine residue and transfers the monophosphate RNA onto a GDP acceptor. A further similarity with that mechanism pertains to the sequence specific nature of the capping reaction – in that substitution of the first or second nucleotides in the pppRNA substrate interferes with the reaction. The present study is very well carried out, likely to be of broad interest and will undoubtedly stimulate a robust effort to target the novel enzymatic activities involved in cap formation in the quest for new antivirals.

Although the study is thorough and well conducted, there are some points that would benefit from being addressed. 1: Previous work has implicated nsp13 as the RNA triphosphatase. The presence of an RNA triphosphatase supports a conventional strategy of mRNA cap formation involving the RTPase and GTase. Here the authors posit that the function of nsp13 is to provide the GDP derived from GTP for the capping reaction. Although they present a reasoned argument to support this model, no experimental tests are carried out. A direct molecular test of this would further strengthen the authors claims. At present although the authors show that provision of radioactive GDP to the covalent nsp9-pRNA resolves to generate an RNA product that is consistent with formation of a capped RNA, they do not show that nsp13 is the source of GDP.

In our assays, $[\alpha\text{-}^{32}\text{P}]\text{-GDP}$ was generated by treating $[\alpha\text{-}^{32}\text{P}]\text{-GTP}$ with nsp13. We apologize for not making this clearer in the manuscript. We have added a sentence to the results section (**lines 140-141**) to emphasize this point. However, the capping reaction does not require the nsp13 protein, as we were able to generate a capped RNA with GDP alone (**See Figure R1a for example**).

We have also performed a kinetic workup of nsp13 NTPase activity. We found that nsp13 displays similar kinetics with ATP and GTP (**Figure R1b, c below and Extended Data Fig. 6 c, d in the revised manuscript**). Mammalian cells contain $\sim 3\text{mM}$ ATP and $500\mu\text{M}$ GTP¹. Therefore, the concentration of ATP and GTP are well above the nsp13 K_m values for both nucleotides.

Figure R1: Nsp13 generates GDP from GTP. **a.** DeRNAylation of nsp9-pRNA^{LS10} by nsp12 in the presence of different NTPs, NDPs or PPi. Reaction products were analysed by SDS-PAGE and Coomassie staining. **b.** Concentration dependence of ATP (blue) or GTP (red) on the rate of phosphate release catalyzed by nsp13. **c.** Table depicting the kinetic parameters of nsp13 with ATP and GTP as substrates.

2: To support the importance of the predicted catalytic residues in cap formation the authors engineer an infectious molecular clone of SARS-CoV-2 to mutate the N-terminal residue of nsp9 N1 to A or D, or N2 to A, and also mutate the catalytic residues K73A and D218A in the NiRAN domain, and D760A of the RdRP domain of nsp12. They assess the effect of those mutations on viral replication using a RT-PCR assay and show that the levels of RNA are reduced 400-4000 fold. The choice of assay here is confusing. One would anticipate that failure to cap the RNA would be lethal for viral replication, rather than resulting in a 2-3 log reduction in viral RNA levels. The viral infectivity should be reported. As viral RNA is produced from the various mutants it would be of significant interest to know whether the RNAs are actually capped or whether they are capped aberrantly. One would also anticipate that production of uncapped RNA would potentially activate antiviral signaling which could readily be measured in the context of the mutants.

The reviewer raises an important point. RT-PCR was probably not the best choice for several reasons, an important one being that it is incredibly sensitive, and these viruses are launched from a plasmid DNA-based system. Even though we use DNaseI in our RT-PCR assay, we can't rule out the possibility that some template DNA gets carried through the experiments and gives low level background in the mutants. Nonetheless, the signal-to-noise for the RT-PCR of WT relative to mutants is very robust. Below is a representative replicate of three newly repeated RT-PCR experiments showing the Ct values, rather than the processed data.

2-NEG	SARS2	36.94
2- WT	SARS2	23.75
2-1NA	SARS2	30.86
2-1ND	SARS2	35.74
2-2NA	SARS2	32.99
2-73KA	SARS2	33.99
2-218DA	SARS2	37.11
2-760DA	SARS2	35.05

We have a 13log₂ dynamic range (NEG 36.9 vs WT 23.8) for the assay and all mutants have Ct values in the 30s, many closer to the NEG sample. Thus, the RNA levels for these samples are just a few-fold above background and likely due to noise in the system. The only exception is nsp9-N1A mutant. As we showed in the manuscript, the nsp9 N1A mutation reduced but did not completely abolish NiRAN-catalyzed NMPylation (**Fig. 1c in the revised manuscript**) or RNAylation (**Fig 2d in the revised manuscript**) of nsp9. Thus, detectable RNA in this sample makes sense. As noted by the reviewer, additional studies that we believe are beyond the scope of this paper are needed to determine the nature of the 5' end of the RNA in the nsp9 N1A mutation and whether antiviral signaling is affected.

The reason we believe these studies are currently beyond the scope of this report is because we took the reviewer's suggestion and performed the more appropriate viral titering assay with WT and all mutants. We could easily detect WT virus by plaque assay, even though the ZsGreen recombinant is attenuated 1-2log₁₀ relative to a non-reporter SARS-CoV-2 (not shown). We could not detect any plaques in undiluted, "neat" supernatants for any of the viral mutants (**Figure R2 below and Fig. 5m in the revised manuscript**). These data confirm that the amino acids examined via mutagenesis are indeed essential for virus production and if mutated, lethal to the virus. Thus, despite some functionality in nsp1-N1A mutant, we propose that the level of RNAylation achieved with this mutant is insufficient to support proper capping,

translation, and launching of viral replication. We have replaced the RT-PCR data with the viral titering data in **Figure 5m** (Also shown in in **Figure R2** below).

Figure R2: The NiRAN domain and the N-terminus of nsp9 are essential for SARS-CoV-2 replication. BHK-21J cells were transfected with SARS-CoV-2-ZsGreen infectious clone DNA. Supernatants were amplified on Vero-C1008-ACE2-TMPRSS2 cells. The Vero supernatants were assayed for infectious virus production by plaque assay. Bars: mean ± SD of N = 3 biological replicates. ND = not detected.

3: Inclusion of quantitative measurements in the main figures would aid the reader – for example, in Figure 1 what is the level of complex formation with nsp9 and the various mutants? Similarly in Figure 2 for the various mutations in the leader RNA sequence. Similarly Figure 5g and h.

We have included quantitative data where appropriate throughout the main figures to aid the reader.

4: Although only a minor point, the main text introduces mRNA cap formation (lines 77-87) in eukaryotic cells. The cap structure was first discovered on viral mRNAs (reovirus and vaccinia) and only subsequently were the details of how the process works in eukaryotes defined. The introduction would benefit from a brief description of this historical context.

We have added a section in the introduction regarding this point. **Lines 75-78 in the revised manuscript.**

Additional minor text issues included

line 37 “how this cap is made is not completely understood”

We have changed the wording to: How this cap is made in SARS-CoV-2 is not completely understood. (We apologize if we misunderstood the issue)

line 54 “topical of these”

We have edited the sentence (**lines 54-55**)

line 61-62 “several promising antivirals used to treat COVID-19 including” Although remdesivir and molnupiravir are used clinically it is arguable as to whether they are promising.

We have edited the sentence (**line 60**)

Referee #3 (Remarks to the Author):

In the submitted manuscript Park and colleagues provide a detailed biochemical characterization of key steps involved in viral RNA capping. They show biochemical evidence that the NiRAN domain of Nsp12 harbors an unexpected RNAylation activity which is crucial

for the formation of GpppA-RNA. The authors suggest the NiRAN domain first catalyzes the transfer of 5'-pRNA to the N-terminus of Nsp9. Subsequently, Nsp12 would facilitate the transfer of 5'-pRNA from Nsp9 to GDP, resulting in GpppA-RNA. Based on this product, the authors successfully recapitulate the formation of the cap-0 and cap-1 structure using recombinant viral proteins Nsp14 and Nsp16, respectively. However, the authors failed to generate a complete high-resolution structure of the E-RTC with native N-terminus of Nsp9. Thus, the authors rely on published data by Yan et al. 2021.

While the biochemical characterizations are well performed, and the manuscript is generally well written, its impact on the scientific community remains questionable. The authors do provide novel insights in the function of the NiRAN domain (RNAylation activity) and the involvement of Nsp9 in RNA capping.

However, other aspects of the manuscript are confirmatory. I recommend this manuscript is considered for transfer to a more specialized Nature journal.

We thank the reviewer for pointing out the novelty of our findings regarding the function of the NiRAN domain. As reviewer 1 and 2 point out, the mechanism of RNA capping in coronaviruses had been a long-standing mystery. Therefore, we believe that our results will have a broad impact on the scientific community, especially given the COVID-19 pandemic.

We would argue that identifying a novel mechanism (kinase-like enzyme mediated RNAylation and GDP-PRNTase activities) that facilitates capping of the SARS-CoV-2 genome and is required for SARS-CoV-2 replication warrants publication in a top tier journal.

We agree that some aspects of our manuscript are confirmatory, and we believe it is important to share these data with the scientific community. For example, we were able to reproduce the NMPylation of nsp9 but were unable to reproduce the NMPylation of nsp7 and nsp8 (**Figure R3, below, Fig. 1b, and Extended data Fig. 3a, k in the revised manuscript**). Likewise, we sought to fully reconstitute the SARS-CoV-2 cap-1 structure, therefore, we performed methylation experiments that confirmed that nsp14 and nsp16 proteins can act on the core cap structure generated by the NiRAN domain (**Figure 4 in the revised manuscript**).

Figure R3. Nsp12 does not NMPylate nsp7 and nsp8 **a.** Incorporation of α -³²P from [α -³²P]ATP, GTP, UTP, or CTP into nsp8 or nsp9 by WT nsp12, the NiRAN mutant (D218A), or the polymerase mutant (D760A). Reactions were performed in the presence of Mg²⁺ or Mn²⁺ and the products were resolved by SDS-PAGE and visualized by Coomassie staining (*top*) and autoradiography (*bottom*). **b.** Incorporation of α -³²P from [α -³²P]GTP or [α -³²P]UTP into nsp7 or nsp9 by WT nsp12, the NiRAN mutant (D218A), or the polymerase mutant (D760A). Reactions were performed in the presence of Mg²⁺ or Mn²⁺ and the products were resolved.

The structure was not intended to be the focus of this paper. The intent of reporting the structure of the RTC bound to nsp9 was to confirm that nsp9 with a native N-terminus occupies a similar position as that reported by Yan et al., which reported a structure with a non-native nsp9 N-terminus². Considering our data identifying the RNylation and GDP-PRNTase activities of the NiRAN domain, we have performed an extensive mutagenesis study of nsp12 that has identified residues involved in both reactions (**Figure R4 below and Figure 5f, g, k, l in the revised manuscript**). We believe this adds to the novelty of our manuscript.

Figure R4. Mutations within the NiRAN domain reduce its capping activity. **a.** Cartoon representation of the NiRAN active site with residues surrounding GDP indicated (PDB: 7CYQ). (**b,c**) Bar graphs representing relative activities of nsp12 mutants in RNylation (**b**) and GDP-PRNTase activities (**c**).

Besides that, several points have to be addressed to increase the clarity.

Line 57: "The SARS-CoV-2 RNA genome contains two open reading frames which are translated by host ribosomes to form two large polyproteins." This sentence is misleading

since it suggests that the genome of SARS-CoV-2 only contains two ORFs, which is obviously not the case.

We have edited the text to read: “The 5' proximal two-thirds of the SARS-CoV-2 RNA genome contains two open reading frames (ORF1a and ORF1ab), which are translated by host ribosomes to form two large polyproteins” (**lines 55-56 in the revised manuscript**)

Line 92: “Furthermore, we present a cryo-EM structure of the SARS-CoV-2 RTC with the native N-terminus of nsp9 bound in the NiRAN active site.” It should be specified here that almost all structural insights are based on previously published data by Yan et al. 2021. The EM-density provided by this study differs only in the most N-terminal amino acids of Nsp9, which are actually crucial for the proposed mechanism. Unfortunately, the authors do state in line 195 that the local resolution of these residues is not sufficient to “distinguish its exact position”. Thus, the EM-densities provided by this study offer limited new insights. Please clarify this in the text.

We have removed this sentence from the introductory paragraph that describes our results. To address this concern, we have also more clearly specified in the results section that our structural analysis of the capping reaction is based mostly on the structure reported by Yan et al.². For example:

(Lines 183-187 in the revised manuscript) “Our cryo-EM analysis was hindered by the preferred orientation of the complex and sample heterogeneity, yielding final maps with high levels of anisotropy, with distal portions of nsp9 missing, and weak density for the N-lobe of the NiRAN domain (Extended Data Fig. 7). Therefore, we mostly use the complex structure by Yan et al.²⁶ (PDBID: 7CYQ) to study the structural basis of NiRAN-mediated RNA capping.”

In the structure by Yan et al.²⁶, Asn1 was assigned an opposite conformation and there are unmodeled residues (non-native N-terminus; NH₂-Gly-Ser-) visible in the density maps, distorting local structural features (Fig. 5c, arrow)³². Nevertheless, the presence of these additional residues does not affect the overall binding mode of nsp9 to the NiRAN domain.

Line 100: “We observed NiRAN-dependent NMPylation of native nsp9, but not native nsp7 or nsp8” and the fact that formation of a GpppA cap was impossible without Nsp9, contradicts published results by Shannon, A. et al. 2021 and Yan, L. et al. 2021, respectively. Some aspects are indeed explainable by the proposed capping pathway of this study. However, it would be good to discuss these aspects in more detail.

We have included a short description in the discussion addressing these contradictions (**lines 272-276 in the revised manuscript**).

In brief, were unable to reproduce the NMPylation of nsp8 (**Figure R3a**) by the NiRAN domain as shown in Shannon et al. 2021³. It is worth noting that Slanina et al.⁴ and Wang et al.⁵ were also unable to NMPylate nsp7 and nsp8. One explanation is that the efficiencies of nsp7 and nsp8 NMPylation are very low (<0.1%), compared to the NMPylation of nsp9 which exhibited nearly 1:1 stoichiometry (**Figure R3c**)

Yan et al.² most likely did not observe NMPylation of nsp9 because they used a non-native N-terminus of nsp9 containing an additional Gly-Ser sequence preceding Asn1, the site of modification.

Line 125 refers to different RNAylation efficiencies depending on the first nucleotide of the RNA substrate and suggests substrate-selectivity. This an interesting point that could be extended to the second nucleotide.

We have extended our analysis and performed experiments testing different nucleotides in the second position. We find that substrate-selectivity is limited to the first nucleotide of the RNA substrate. (Figure R5 below and Fig. 2f, g in the revised manuscript).

Figure R5. NiRAN-mediated RNAylation is specific for the first nucleotide of the SARS-CoV-2 leader sequence. **a.** Incorporation of RNAs with substitutions in the first and second base into nsp9 by the NiRAN domain. Reaction products were analysed by SDS-PAGE and Coomassie staining. **b.** Quantification of RNAylation. Results shown are from three independent experiments. Error bars represent the standard deviation.

Throughout the manuscript the authors extensively test the influence of Mn²⁺ and Mg²⁺ on biochemical reactions. Accordingly, it would be interesting to better discuss the role of different divalent ions during the capping reactions.

This is a good question. Unfortunately, we do not have a satisfactory answer as to why different divalent cations are preferred. However, it is worth noting that different kinases prefer different divalent cations to catalyze phosphorylation.

Line 205 mentions a mutant version of Nsp12 that just contains the NiRAN domain (Δ RdRp; 1-326). This mutant does not show any signs of enzymatic activity, which is presented as conformation for the described interactions between Nsp12 and Nsp9. However, no other biochemical data which characterizes the Δ RdRp mutant is shown. So it remains unclear if the deletion of such a large domain results in a properly folded protein. This must at least be discussed.

We believe that the Δ RdRp protein is folded properly given that it runs as a single peak at its predicted molecular weight based on size exclusion chromatography (SEC) (Figure R6a below and Extended data Fig. 8a in the revised manuscript). Our intention with the Δ RdRp mutant was to assess whether there may be additional residues from the RdRp domain which can mediate the capping reaction. To better address this question, we generated a point mutation in a conserved arginine (R733) within the RdRp domain that positions Asn2 of nsp9

near the active site of the NiRAN domain (**Figure R6b below and Fig. 5e in the revised manuscript**). Alanine substitution of R733 markedly reduced RNAylation and PRNTase activities (**Figure R6c-d below and Fig. 5f, g in the revised manuscript**).

Figure R6: The RdRp domain is required for NiRAN mediated capping of RNA. **a.** SEC analysis of the Δ RdRp mutant. Standards are shown. **b.** Cartoon depiction of R733 from the RdRp domain (magenta) interacting with Asn2 on nsp9 (gold) near the NiRAN active site. **c, d.** Relative RNAylation (**c**) and GDP-PRNTase (**d**) activities of WT and R733A nsp12.

Figure 3d suggests that deRNAylation of Nsp9 is specific for GDP but also to a significant extent for GTP. However, in all following reactions involving Nsp12 and Nsp9 the effect of GTP was not tested. An important claim of this manuscript is that Nsp13 is essential for the proposed capping reaction since it catalyzes the formation of GDP (an activity Nsp12 lacks). Thus it would be interesting to see if the NiRAN domain is indeed incapable of utilizing GTP as substrate for the formation of GpppA-RNA.

We have repeated the capping assays using [α - 32 P]GTP and [α - 32 P]GDP and separated the reaction products by TLC to assess the formation of GpppA-RNA. We do observe formation of GpppA using both GTP and GDP, although the reaction was more efficient with GDP (compare GDP at 15 minutes vs. GTP at 15 minutes) (**Figure R7a below, and Extended Data Fig. 10a in the revised manuscript**). However, we found that the “[α - 32 P]GTP” sample from Perkin Elmer that is used in our reactions does have significant amounts of GDP, likely as a result of spontaneous hydrolysis of the GTP (**Figure R7b below, and Extended Data Fig. 10b in the revised manuscript**). We also did not detect any GppppA (G(p)₄A), which would be expected to form if the NiRAN domain could use GTP in these assays. Thus, the GpppA that is formed in the presence of GTP is generated from GDP in the sample. This is also likely why we see some deRNAylation of nsp9 in the presence of GTP (**Figure R7c below and Fig. 3d in the revised manuscript**).

Figure R7 The NiRAN domain is specific for GDP in the capping reaction. **a.** TLC analysis of the reaction products following incubation of nsp12 and nsp9-pRNA^{L510} with [α -³²P]GTP or [α -³²P]GDP. **b.** TLC analysis of [α -³²P]GTP or [α -³²P]GDP used in the assays in **a.** Note the presence of GDP in the GTP sample. **c.** DeRNAylation of nsp9-pRNA^{L510} by nsp12 in the presence of different NTPs, NDPs or PP_i. Reaction products were analysed by SDS-PAGE and Coomassie staining.

Figure 5j presents intriguing data which suggests that kinase-like residues of the NiRAN domain and the N-terminus of nsp9 are essential for SARS-CoV-2 replication. Unfortunately, relative viral yields of duplicates instead of triplicates are depicted. This results in very large error bars, making it impossible to differentiate between the individual effects. This must be clarified.

As mentioned in response to Reviewer 2, point 2:

RT-PCR was probably not the best choice for several reasons, an important one being that it is incredibly sensitive, and these viruses are launched from a plasmid DNA-based system. Even though we use DNaseI in our RT-PCR assay, we can't rule out the possibility that some template DNA gets carried through the experiments and gives low level background in the mutants. Nonetheless, the signal-to-noise for the RT-PCR of WT relative to mutants is very robust. Below is a representative replicate of three newly repeated RT-PCR experiments showing the Ct values, rather than the processed data.

2-NEG	SARS2	36.94
2-WT	SARS2	23.75
2-1NA	SARS2	30.86
2-1ND	SARS2	35.74
2-2NA	SARS2	32.99
2-73KA	SARS2	33.99
2-218DA	SARS2	37.11
2-760DA	SARS2	35.05

We have a 13log₂ dynamic range (NEG 36.9 vs WT 23.8) for the assay and all mutants have Ct values in the 30s, many closer to the NEG sample. Thus, the RNA levels for these samples are just a few-fold above background and likely due to noise in the system. The only exception is nsp9-N1A mutant. As we showed in the manuscript, the nsp9 N1A mutation reduced but did not completely abolish NiRAN-catalyzed NMPylation (**Fig. 1c in the revised manuscript**) or RNylation (**Fig. 2d in the revised manuscript**) of nsp9. Thus, detectable RNA in this sample makes sense.

We took reviewer 2's suggestion and performed the more appropriate viral titering assay with WT and all mutants. We could easily detect WT virus by plaque assay, even though the ZsGreen recombinant is attenuated 1-2log₁₀ relative to a non-reporter SARS-CoV-2 (not shown). We could not detect any plaques in undiluted, "neat" supernatants for any of the viral mutants. These data confirm that the amino acids examined via mutagenesis are indeed essential for virus production and if mutated, lethal to the virus. Thus, despite some functionality in nsp1-N1A mutant, we propose that the level of RNylation achieved with this mutant is insufficient to support proper capping, translation, and launching of viral replication. We have replaced the RT-PCR data with the viral titering data in **Figure 5m** (Also shown in **Figure R8 below**). We also show newly generated triplicate RT-PCR data below in **Figure R8b below and Extended Data Fig. 9b in the revised manuscript**; note that this data is not on a log scale like the original data)

Figure R8: The NiRAN domain and the N-terminus of nsp9 are essential for SARS-CoV-2 replication. (a, b) BHK-21J cells were transfected with SARS-CoV-2-ZsGreen infectious clone DNA. Supernatants were amplified on Vero-C1008-ACE2-TMPRSS2 cells. The Vero supernatants were assayed for infectious virus production by plaque assay (a) or for viral RNA levels by RT-qPCR (b). Bars: mean \pm SD of N = 3 biological replicates. ND = not detected.

References

- 1 Traut, T. W. Physiological concentrations of purines and pyrimidines. *Mol Cell Biochem* **140**, 1-22, doi:10.1007/BF00928361 (1994).
- 2 Yan, L. *et al.* Cryo-EM Structure of an Extended SARS-CoV-2 Replication and Transcription Complex Reveals an Intermediate State in Cap Synthesis. *Cell* **184**, 184-193 e110, doi:10.1016/j.cell.2020.11.016 (2021).
- 3 Shannon, A. *et al.* Protein-primed RNA synthesis in SARS-CoVs and structural basis for inhibition by AT-527. *bioRxiv*, 2021.2003.2023.436564, doi:10.1101/2021.03.23.436564 (2021).

- 4 Slanina, H. *et al.* Coronavirus replication-transcription complex: Vital and selective NMPylation of a conserved site in nsp9 by the NiRAN-RdRp subunit. *Proc Natl Acad Sci U S A* **118**, doi:10.1073/pnas.2022310118 (2021).
- 5 Wang, B., Svetlov, D. & Artsimovitch, I. NMPylation and de-NMPylation of SARS-CoV-2 nsp9 by the NiRAN domain. *Nucleic Acids Res* **49**, 8822-8835, doi:10.1093/nar/gkab677 (2021).

Reviewer Reports on the First Revision:

Referees' comments:

Referee #2 (Remarks to the Author):

The authors have addressed my comments. In particular it is not at all surprising to see that infectious virus cannot be recovered from the mutants in the capping enzyme.

Referee #3 (Remarks to the Author):

The authors have addressed the concerns and I recommend publication. It is up to the editor to decide whether Nature or another Nature group journal is the right choice.

Referee #4 (Remarks to the Author):

This is an interesting paper that examines the mechanism of 5'-capping by SARS CoV-2. As for nearly all RNA viruses, capping is important for viral replication, enabling transcripts to hijack the cellular translation machinery. Viruses utilize a variety of mechanisms to perform 5'-end capping, and in this case, the function is achieved by viral rather than cellular enzymes. The work is important and timely and should be of broad interest.

I paid particular attention to the NMR studies since these are most closely aligned with my area of expertise. The ³¹P-correlated experiments definitively demonstrate that AMP is covalently attached to the protein via a phosphodiester linkage. I recommend that Extended Figure 4 be included in the main text, if possible, as it confirms that AMP is linked to the backbone rather than a side chain. The TOCSY experiments alone don't distinguish between backbone and a possible (but unlikely) side-chain linkage, and the data in Extended Figure 4 are truly beautiful).

Author Rebuttals to First Revision:

Referees' comments:

Referee #2 (Remarks to the Author):

The authors have addressed my comments. In particular it is not at all surprising to see that infectious virus cannot be recovered from the mutants in the capping enzyme.

Referee #3 (Remarks to the Author):

The authors have addressed the concerns and I recommend publication. It is up to the editor to decide whether Nature or another Nature group journal is the right choice.

Referee #4 (Remarks to the Author):

This is an interesting paper that examines the mechanism of 5'-capping by SARS CoV-2. As for nearly all RNA viruses, capping is important for viral replication, enabling transcripts to hijack the cellular translation machinery. Viruses utilize a variety of mechanisms to perform 5'-end capping, and in this case, the function is achieved by viral rather than cellular enzymes. The work is important and timely and should be of broad interest.

I paid particular attention to the NMR studies since these are most closely aligned with my area of expertise. The ³¹P-correlated experiments definitively demonstrate that AMP is covalently attached to the protein via a phosphodiester linkage. I recommend that Extended Figure 4 be included in the main text, if possible, as it confirms that AMP is linked to the backbone rather than a side chain. The TOCSY experiments alone don't distinguish between backbone and a possible (but unlikely) side-chain linkage, and the data in Extended Figure 4 are truly beautiful).

We thank the reviewer for the positive comments regarding our manuscript and the NMR results. As requested, we have moved the NMR data in Extended Data 4 to Main Fig. 1.